



# Quantifying forecast uncertainty of Mediterranean cyclone-related surface weather extremes in ECMWF ensemble forecasts. Part 1: Method and case studies

Katharina Hartmuth[1], Dominik Büeler[1,a], and Heini Wernli[1]

[1]Institute for Atmospheric and Climate Science, ETH Zurich, Zurich, Switzerland
[a]now at: Federal Office of Meteorology and Climatology, MeteoSwiss, Zurich, Switzerland

**Correspondence:** Katharina Hartmuth (katharina.hartmuth@env.ethz.ch)

**Abstract.** Extratropical cyclones are the main cause of high-impact weather events in the Mediterranean such as heavy precipitation, floods, severe winds, and dust storms. However, the accuracy in predicting the timing, location, and intensity of such events is often insufficient, which is typically related to errors in cyclone position, propagation, and intensity. In this two-part study we use operational forecasts from the ECMWF ensemble prediction system to quantify uncertainties in predicting high-impact weather conditions linked to Mediterranean cyclones. We apply an object-based approach to attribute Mediterranean cyclones to events of extreme precipitation and surface winds. In this first part, we introduce the probabilistic method and three illustrative case studies of Mediterranean cyclones that occurred between November 2022 and September 2023, including the infamous Storm Daniel as well as Storms Denise and Jan. We find that the cyclones as well as their attributed objects of extreme surface weather are predicted well for lead times $\leq 48\,\text{h}$. However, for longer lead times there is large case-to-case variability in the ensemble performance. Predictions of extreme surface weather objects are found to be more uncertain (i) for smaller and less coherent objects, (ii) if the attributed cyclone is captured by fewer ensemble members, and (iii) during the earlier stage of the cyclones' lifecycle.

## 1 Introduction

Early on, the Mediterranean has been identified as an active cyclogenetic region with hot spots of cyclone formation in the Gulf of Genoa in the western and near Cyprus in the eastern Mediterranean, respectively (e.g., Gleeson, 1953; Pettersen, 1956). Mediterranean cyclones develop mostly due to baroclinic instability with their evolution often being affected by the complex topography surrounding the Mediterranean Sea and by diabatic processes (Trigo et al., 1999; Flaounas et al., 2022). Their occurrence has a significant seasonal cycle with a notable increase in more intense and longer-lived cyclones in winter (Lionello et al., 2006; Campins et al., 2011; Flaounas et al., 2018). Although Mediterranean cyclones are typically weaker, smaller, and shorter-lived than extratropical cyclones in the main storm track regions (Trigo, 2006; Čampa and Wernli, 2012; Flaounas et al., 2014), they are the major cause of high-impact weather events in the Mediterranean, including heavy precipitation and floods, severe winds and dust storms (e.g., Pfahl and Wernli, 2012; Raveh-Rubin and Wernli, 2015; Flaounas et al., 2022). Jansà et al. (2001) found that over 90 % of heavy rainfall events in the western Mediterranean are attributed to cyclones,





and surface wind extremes are found to be almost always related to a cyclone (Nissen et al., 2010; Raveh-Rubin and Wernli,
2015), as are compound precipitation-wind extremes (Raveh-Rubin and Wernli, 2015; Portal et al., 2024). Due to the dense
population of the Mediterranean region, cyclones that are attributed to extreme surface weather can cause severe environmental
and socio-economic damage (e.g., EUMETSAT, 2018; FloodList, 2022; WMO, 2023; Khodayar et al., 2025).

To adequately prepare for extreme Mediterranean weather, improve early-warning systems and issue measures such as
evacuation procedures in time, it is essential to evaluate recent versions of forecast models. The predictability of Mediter-
ranean cyclones in the ensemble forecast and reforecast products of the European Centre for Medium-Range Weather Fore-
casts (ECMWF) has been assessed in several studies, concluding that Mediterranean cyclones can be accurately predicted by
the ensemble, but only few days in advance (Pantillon et al., 2017; Di Muzio et al., 2019; Vollenweider, 2023). For longer
lead times, a systematic underestimation of cyclone speed and intensity has been found (Pantillon et al., 2017). Furthermore,
improved forecasts of the cyclone location were found to be linked to slower and more shallow cyclones (Doiteau et al., 2024).
This agrees with the findings of Froude et al. (2007a), who further showed that in general, the forecast skill for the position
of extratropical cyclones is significantly higher than for their intensity. Despite the poor forecast skill for rapidly intensifying
cyclones and the resulting less accurate prediction of winter cyclones in the Mediterranean, Doiteau et al. (2024) could show
that the more intense a cyclone is, the more ensemble members actually detect this cyclone compared to shallower cyclones.
Furthermore, Di Muzio et al. (2019) showed that ensemble forecasts typically cannot predict medicanes, a category of partic-
ularly severe Mediterranean cyclones (e.g., Fita et al., 2007; Di Muzio et al., 2019), further ahead than 5 to 7 d, a result that is
consistent with the existence of predictability barriers (Riemer and Jones, 2014; Pantillon et al., 2016). They also revealed that
forecasts initialized after the formation of the cyclone are distinctively more accurate than earlier forecasts, which still need to
capture the cyclogenesis process.


The predictability of cyclone-attributed extreme rainfall and surface winds varies substantially from case to case (e.g., Pan-
tillon et al., 2017). Several studies showed that the forecast performance is sensitive to both the structure and amplitude of
upper-level potential vorticity (PV) anomalies as well as to the interaction of the low-level flow with the complex Mediter-
ranean topography and, in particular, to the position and intensity of the surface cyclone (Fehlmann and Quadri, 2000; Romero
et al., 2005; Argence et al., 2009; Horvath and Ivančan-Picek, 2009). Simulating a heavy rainfall event, Argence et al. (2009)
found that initial perturbations applied to an upper-level trough intensified throughout the simulation, resulting in forecast
discrepancies of the low-level cyclone. They further showed that the overall precipitation pattern was controlled by both the
upper-level PV structure and the cyclone at the surface. However, the predictability of smaller-scale features such as localized
heavy precipitation was found to be directly related to the forecast of the location and/or intensity of the surface low (Romero
et al., 2005).

While several studies investigated the predictability of cyclones in the main storm tracks (e.g., Froude et al., 2007b; Froude,
2010; Zheng et al., 2017; Korfe and Colle, 2018), the predictability of Mediterranean storms has been studied to a lesser extent.



Despite the increasing knowledge about factors affecting the predictability of extreme surface weather linked to Mediterranean
cyclones, forecasting such events, in particular their timing, location, and intensity, has often been found to be insufficiently
accurate (e.g., Davolio et al., 2015). Events such as the catastrophic flooding of the city of Derna, Libya, attributed to Storm
Daniel in September 2023 (e.g., CBS News, 2023; Greek Reporter, 2023; WMO, 2023; Armon et al., 2025) emphasize the
relevance of such storms for infrastructure and human safety in the region and the socio-economic need for accurate predictions
and warnings.


This two-part study investigates the prediction of high-impact weather conditions linked to Mediterranean cyclones in the
operational ECMWF ensemble prediction system (ENS). Our overall goals are (1) to develop a method that allows quantifying
the probabilities of predicting extreme surface wind and precipitation related to Mediterranean cyclones at different lead times,
(2) to illustrate the method with case studies and later compile a multi-year statistical analysis of a large set of Mediterranean
cyclones, (3) to assess whether the forecast uncertainty of surface weather extremes depends on cyclone characteristics such
as their position, intensity, and propagation speed, and finally (4) to investigate whether forecast uncertainty of high-impact
weather conditions in the Mediterranean is influenced by upstream processes over the North Atlantic, e.g., a warm conveyor
belt influencing Rossby wave breaking over Europe (Raveh-Rubin and Flaounas, 2017; Scherrmann et al., 2024). Address-
ing these objectives and visualizing the results is conceptually and technically challenging, in particular because of the large
amount of data and analyses required. For instance, for a single cyclone in ERA5, considering the corresponding ensemble
forecasts twice daily with 50 members each and lead times up to 10 d, requires the identification of the cyclone and its at-
tributed extreme weather objects in 1000 individual forecasts. Therefore, in this first part of the study, we mainly introduce
our method to identify Mediterranean cyclone-related surface weather extremes in ENS and illustrate the method with three
contrasting case studies. Thereby, this first part addresses our research objective (1) and parts of (2) and (3). Results from the
statistical multi-year investigation will be presented in part two of this study.

This paper is organized as follows: We present the datasets and methods used in Sect. 2, followed by an introduction of three
case study cyclones and their representation in ERA5 in Sect. 3. In Sect. 4, we analyze the performance of ENS, in particular
the representation of the cyclone track and attributed objects of extreme surface weather in the forecasts at different lead times.
We discuss and conclude our results in Sect. 5.

## 2   Data and method

This study uses two different datasets from the European Centre of Medium-Range Weather Forecasts (ECMWF), namely
the ERA5 reanalysis and medium-range ensemble forecasts (ENS) produced with the ECMWF Integrated Forecasting System
(IFS), which are briefly introduced in Sect. 2.1 and 2.2. Then we present the methodological steps required to address the
research questions outlined in the introduction. These steps are:





1. Identification and tracking of Mediterranean cyclones as 2-dimensional objects in ENS, and matching with cyclones in ERA5 (Sect. 2.3).

2. Identification of 2-dimensional objects of extreme surface precipitation and 10-m wind gusts (Sect. 2.4).

3. Attribution of extreme surface weather objects to cyclone objects (Sect. 2.5).

4. Probabilistic analysis of extreme surface weather objects relative to cyclone center (Sect. 2.6).

## 2.1 ERA5 reanalysis

The ERA5 reanalysis (Hersbach et al., 2020) is used as a reference dataset for forecast validation, and interpolated to a grid with a spatial resolution of 0.5°x0.5° and a 1-hourly temporal resolution. Reduced sea level pressure (SLP) is used to identify cyclones, while fields of total precipitation ($P$) and 10 m wind gusts ($G_{10}$) are utilized to identify objects of extreme surface weather. A 30-year climatology from 1990–2019 is considered to define thresholds for extreme events (see Sect. 2.4). Furthermore, in Sect. 3, the large-scale environment of the case study cyclones is investigated with fields of potential temperature ($\theta$) at 850 hPa and potential vorticity (PV) on isentropic levels.

## 2.2 Operational IFS ensemble forecasts (ENS)

Twice per day, at 00 and 12 UTC, the ECMWF runs 50 ensemble members with slightly perturbed initial conditions for their medium range ensemble forecast. Since May 2022, we have been retrieving several fields from these forecasts quasi-operationally with full vertical resolution on the 68 lowest model levels in a domain covering North America, Greenland, the North Atlantic, Northern Africa, and Europe, on a horizontal grid with 0.5° grid spacing[1]. The active model versions since starting the download include IFS Cycles 47r3, 48r1 and 49r1. Due to the substantial storage capacities that would be needed to keep this data, many of the subsequently described object-based postprocessing steps are also performed quasi-operationally twice a day before archiving parts of the data on our servers and eventually on tape. Nevertheless, a selection of atmospheric fields is stored, including temperature, wind, moisture and precipitation, and additionally calculated secondary parameters such as $\theta$ and PV. For each ensemble member, data is downloaded every 6 h up to a maximum lead time of 15 d. This continuously growing dataset allows us to assess current prediction capabilities of extreme weather events and associated dynamical features such as surface cyclones in the North Atlantic European sector for a multi-year period.

## 2.3 Cyclone identification, tracking, and cyclone track matching

Extratropical cyclones are identified in ERA5 and in each ensemble member of ENS as two-dimensional objects (Wernli and Schwierz, 2006). Thereby, a cyclone is defined as the area around a local minimum in SLP, bounded by the outermost closed SLP contour. To identify cyclone tracks, the algorithm presented in Sprenger et al. (2017) is applied. A cyclone track has to

---

[1]Note that ECMWF only archives 3-dimensional variables from ENS on selected pressure levels; we therefore transfer the required data on model levels soon after forecast completion, before the operational archiving occurs at ECMWF.



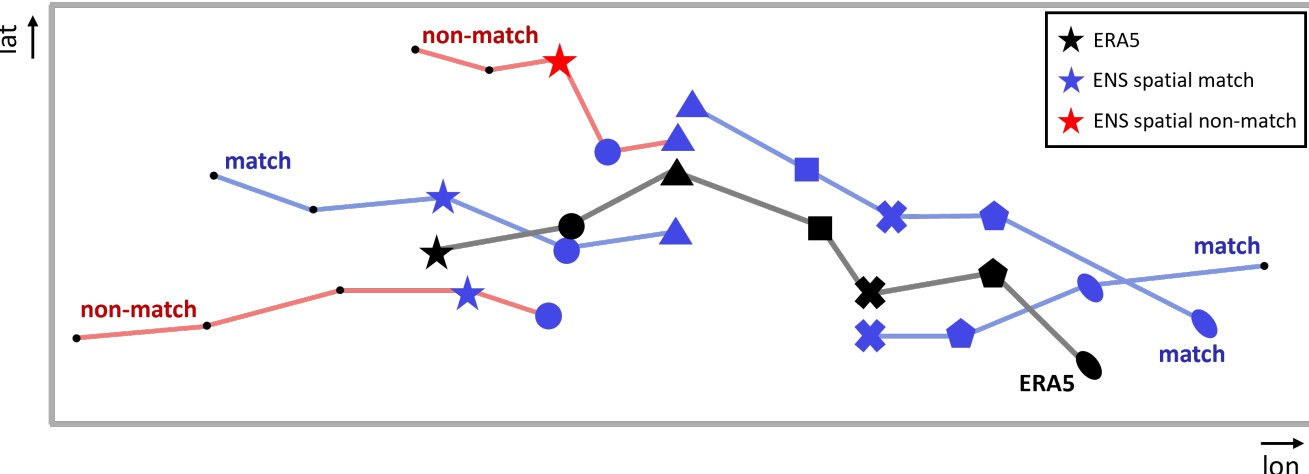

**Figure 1.** Schematic of the cyclone track matching algorithm. Black line denotes a cyclone track in ERA5, colored lines denote ensemble member tracks. Identical symbols refer to the same 6-hourly timestep. Blue symbols and lines denote time steps and tracks, respectively, that match the spatial criterion, and red symbols and lines situations that do not match the spatial criterion.

exist for at least 24 h to be considered. Mediterranean cyclones are identified as cyclones that reach their mature stage, i.e.,
their minimum central SLP, within a "Mediterranean box" extending from 10°W to 40°E and 30°N to 47°N (except for the Bay of Biscay in the northwestern corner). Note that to make both datasets comparable, cyclone tracks in ERA5 (which are based on hourly SLP fields) are only considered every 6 h, i.e., at 00, 06, 12 and 18 UTC.

To investigate the ability of the ENS members to predict an observed cyclone track as identified in ERA5, a cyclone track
matching algorithm is applied for Mediterranean cyclones similar to Flaounas et al. (2023). For each Mediterranean cyclone track in ERA5, the following criteria are applied to identify matching cyclone tracks in each ENS member:

- *Spatial criterion:* At a certain time step, the location of the cyclone center in an ensemble member has to be within a radius of 300 km around the cyclone center in ERA5 to be regarded as a match.

- *Temporal criterion:* For a cyclone track in an ensemble member to be matched with an ERA5 cyclone track, the spatial
criterion has to be fulfilled for at least three consecutive time steps, that is a minimum of 12 h. Note that this period is shorter than the one used by Flaounas et al. (2023), who applied this method to compare different tracking methods within the identical dataset as opposed to the matching of cyclone tracks between reanalyses and forecasts in our case.

With this choice of criteria, the number of matching tracks per time step along the ERA5 cyclone track can vary considerably, particularly for longer cyclone tracks. It is further possible that more than one matching cyclone track is found in a single
ensemble member, either because of a gap in the identified cyclone track or due to the presence of two cyclones in the forecast, one matching the ERA5 track in the earlier stage and the second one in the later stage of the ERA5 cyclone. To avoid the





occurrence of two matching tracks in the same ensemble member at the same time step, only the ensemble member track point closest to the ERA5 track is considered.

## 2.4 Definition of extreme surface weather objects

To investigate the performance of ENS in predicting extreme surface weather attributed to the identified cyclones, we define two-dimensional objects of extreme precipitation and extreme surface wind gusts as follows. First, a threshold is determined at each grid point based on the 99th percentile of $P$ and $G_{10}$ in the ERA5 reanalysis dataset. The percentiles are determined separately for each season, i.e., for December to February, etc. Although it would be preferable to similarly use an ENS-based percentile threshold for the analysis of extreme weather objects in ENS, this is not practical given the challenge to obtain

an operational IFS model climatology without downloading an extensive amount of additional hindcast data. Therefore, we pragmatically use the threshold defined in ERA5 also for ENS. In a next step, adjacent grid points that exceed such a threshold are defined as extreme surface weather objects described by a two-dimensional binary field with a value of 1 at grid points inside the object, and 0 outside. In this way, coherent regions of intense $P$ and $G_{10}$ are objectively identified in the ERA5 reanalysis as well as in each ENS member. In the following, we will refer to such objects as "extreme objects".

## 150 2.5 Attribution of extreme objects to cyclones

Extreme objects are linked to surface cyclones based on the following objective approach: If at a certain time step an extreme object overlaps with the circle spanned by a radius of $400\,\mathrm{km}$ around the cyclone center, the entire extreme object is attributed to the cyclone. In a study by Portal et al. (2024), such a threshold distance was found to be reasonable to identify precipitation extremes attributed to surface cyclones in the Mediterranean. This study further showed that surface wind extremes linked to a

surface cyclone usually extend to areas more distant from the cyclone center. However, to avoid an overlap of extreme objects attributed to different cyclones, we choose to apply the same threshold for both parameters.

## 2.6 Probability of extreme objects

To assess the performance of ENS in forecasting extreme objects relative to a surface cyclone, a cyclone-centered probability of such objects in ENS for a given forecast and lead time is calculated at each time step along the ERA5 track, as shown in

Fig. 2. In a first step, for each ensemble member, all extreme objects within a box of $\pm 10°$ relative to the cyclone center as represented in this ensemble member are collected (illustrated exemplarily for five ensemble members in Fig. 2a-e). A value of 1 is attributed to all grid points within an extreme object. Members without a matching cyclone contribute with a constant field of 0 to the overall probability. In a second step, taking the mean over all 50 ensemble members results in the probability of an extreme object near the cyclone center in ENS (Fig. 2g,h). This probability is referred to as *unconditional probability* and

used throughout the manuscript unless otherwise stated. In addition, a *conditional probability* is computed in the same way, but considering only ensemble members with a matching cyclone instead of all members. Such conditional probabilities provide



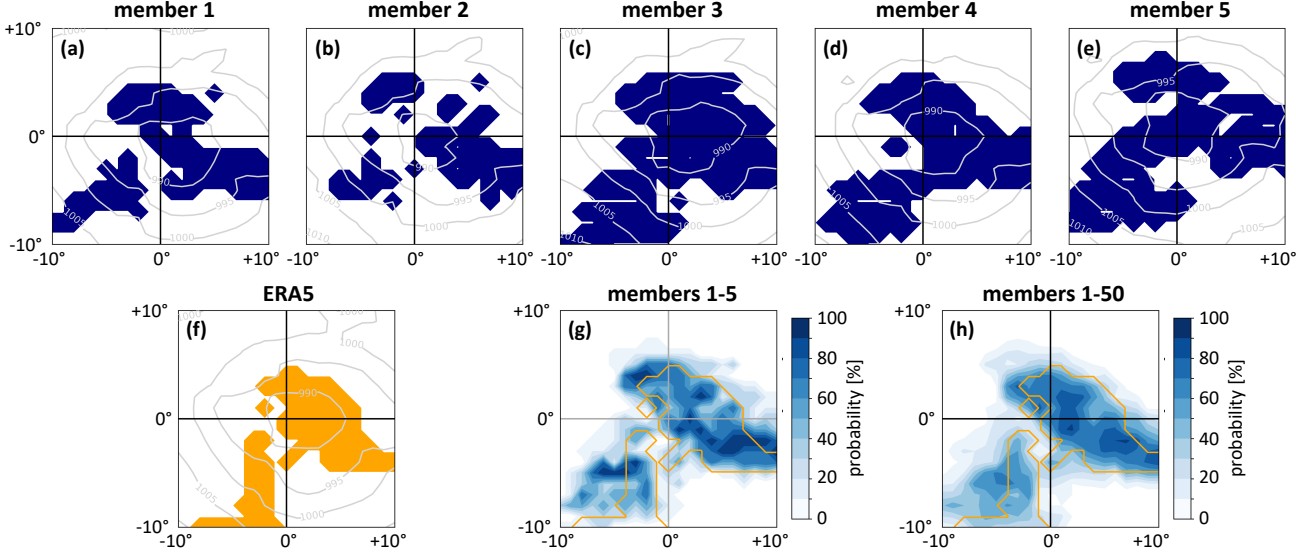

**Figure 2.** Illustration of the calculation of cyclone-centered probabilities of extreme objects in ENS. **(a-e)** show cyclone-centered extreme objects (dark blue shading) and SLP fields (grey contours; in intervals of 5 hPa) for five randomly selected members; **(f)** shows the extreme object (orange) and SLP field at same time step in ERA5. **(g)** depicts the average of (a-e), i.e., the object probability for the five ENS members (blue shading; in %) and the ERA5 object (orange contour). (h) shows object probability including all 50 ENS members.

insight about the ability of the model to forecast extreme objects given that a matching surface cyclone exists.

In this paper, three case studies are performed to illustrate our methodology of quantifying the performance of ENS in
forecasting extreme surface weather attributed to Mediterranean cyclones. The case studies are chosen based on the occurrence of such extreme weather events since the start of our systematic ENS evaluation in May 2022. Also, the cases are chosen such that they represent different types of Mediterranean cyclones, which helps testing the methodology across the spectrum of cyclones with different characteristics. Key characteristics of the cases are summarized in Table 1. Case study 1 (Storm Denise) investigates a classical lee cyclone in the Gulf of Genoa (Tafferner, 1990; Buzzi et al., 2020); case study 2 (Storm Jan)
a cyclone that propagates from the North Atlantic into the western and central Mediterranean, with a track similar to Storm Klaus (Liberato et al., 2011); and case study 3 (Storm Daniel) an exceptionally long-lived cyclone in the eastern Mediterranean that developed medicane characteristics (Flaounas et al., 2024). First, we introduce the three storms using ERA5 reanalysis data before analyzing their representation in ENS.



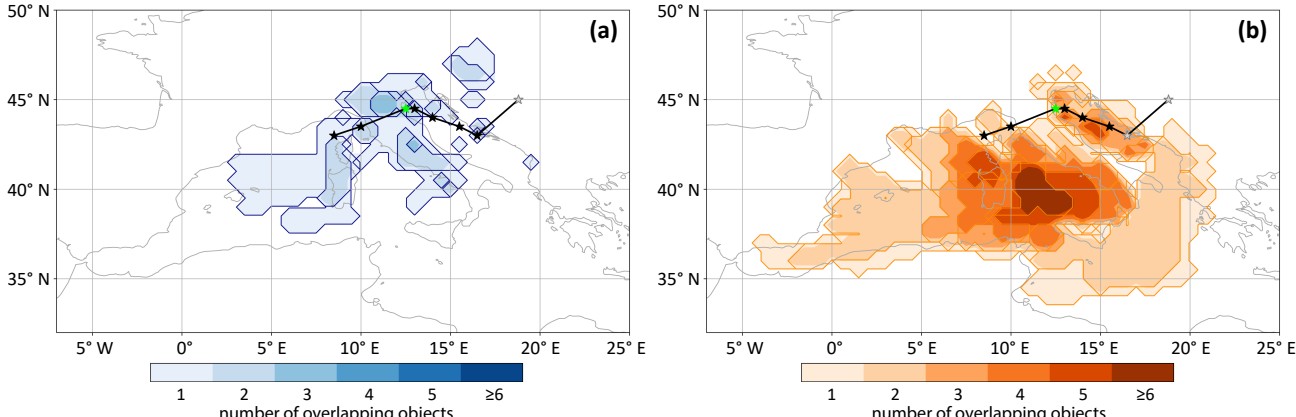

**Figure 3.** Track of Storm Denise (black line) with 6-hourly objects of extreme **(a)** precipitation and **(b)** surface wind gusts (shading) in ERA5. Single track points are denoted every 6 h by black stars if they are attributed to an object and by grey stars if no object occurs at that time step. Green star marks time and location of maximum cyclone intensity (minimum SLP).

## 3 Case study overview

| Storm | Genesis time | Lifetime [h] | $SLP_{min}$ [hPa] | Maximum intensification [hPa] | Area extreme $P$ [$10^5\,\mathrm{km}^2$] | Area extreme $G_{10}$ [$10^5\,\mathrm{km}^2$] |
|---|---|---|---|---|---|---|
| **Denise** | 22 Nov 2022 00 UTC | 42 | 985.4 | −6.6 | 6.01 | 36.36 |
| **Jan** | 18 Jan 2023 18 UTC | 78 | 991.0 | −10.0 | 6.56 | 10.82 |
| **Daniel** | 04 Sep 2023 12 UTC | 174 | 995.6 | −7.1 | 24.92 | 7.18 |

**Table 1.** Characteristics of all three case study storms, including their genesis time, lifetime, minimum SLP ($SLP_{min}$), their maximum intensification (SLP decrease in 12 h for Denise and 18 h for the two other cases), and the total affected area of $P$ and $G_{10}$ extremes, respectively, integrated along the cyclone track.

### 3.1 Case study 1: Storm Denise – November 2022

The first case study investigates Storm Denise, which formed at 00 UTC on 22 November in the Gulf of Genoa, causing severe wind gusts, high waves and storm surge over Mallorca and Corsica (Majorca Daily Bulletin, 2022; European Severe Weather Base, 2022). During 22 November it passed over northern Italy where it reached its minimum SLP of less than 990 hPa at 12 UTC, leading to strong winds and heavy precipitation mainly in the Emilia-Romagna region (FloodList, 2022) before reaching the coast of Croatia on 23 November, where extreme winds were observed along the Adriatic coast (European Severe Weather Base, 2022). Several fatalities were reported in Italy, mainly related to a landslide that was partially triggered after the passage of the storm (Agenzia Italia, 2022; CNN World, 2022). Cyclolysis already happens 42 h after genesis, such that this is





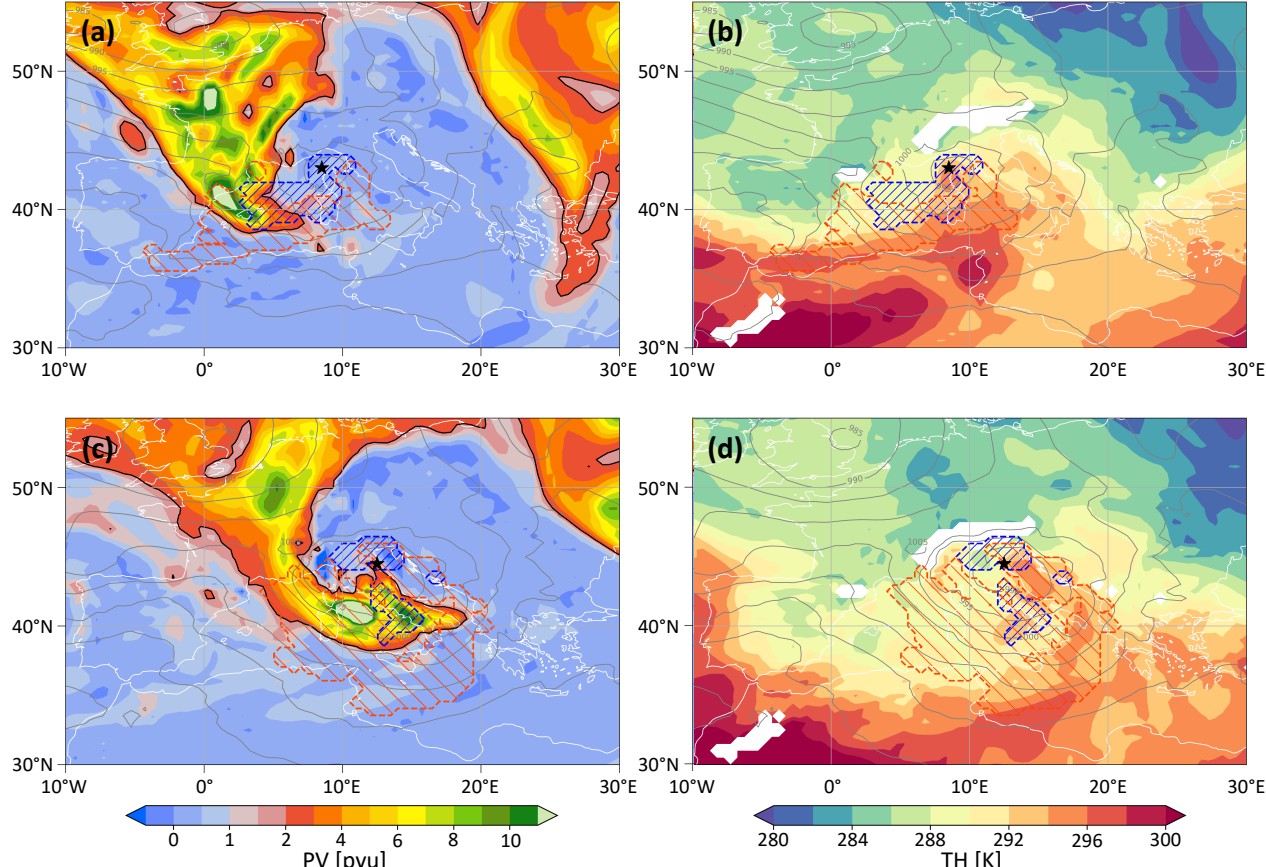

**Figure 4.** Maps of **(a, c)** PV on 315 K (color; in pvu) and **(b, d)** $\theta$ at 850 hPa (color; in K) at (a, b) 00 UTC on 22 November (time of cyclogenesis) and (c, d) 12 UTC on 22 November (time of minimum SLP, 12 h after genesis). Cyclone center is marked by black star. Areas of extreme precipitation and surface wind gusts are shown with blue and orange hatching, respectively. Grey lines denote SLP in intervals of 5 hPa.

the cyclone with the shortest lifetime but lowest SLP minimum among the three cases (Table 1).

Associated areas of extreme $P$ and $G_{10}$ in the ERA5 dataset are shown in Fig. 3. Colors do not indicate the intensity of the extremes, but their duration at a given grid point. For instance, a number of 3 overlapping objects indicates that at this location the extreme conditions attributed to the same cyclone persisted for 18 h (three 6-hourly time steps). While extreme $P$ occurred mainly during the intensification stage of the cyclone (Fig. 3a), attributed extreme $G_{10}$ affected large parts of the Mediterranean Sea and especially Italy and Corsica, covering the largest areas near the time of the cyclone's mature stage at 12 UTC on 22



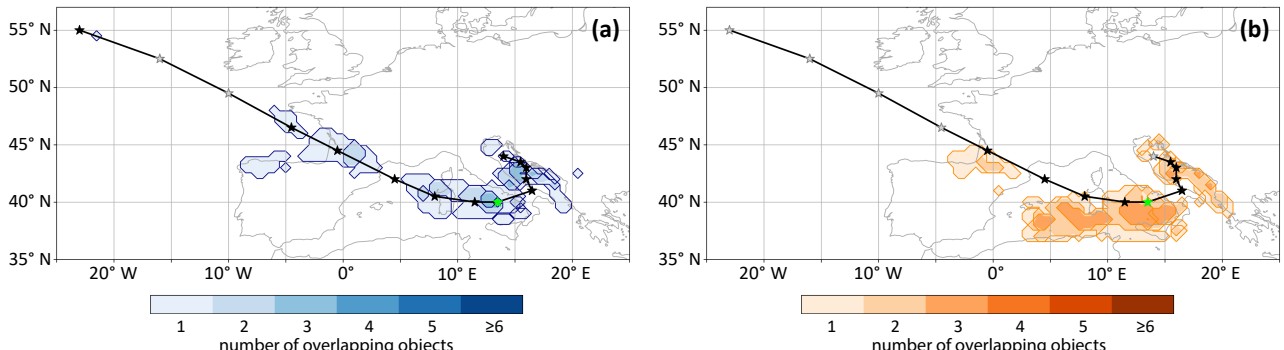

**Figure 5.** Same as Fig. 3 but for Storm Jan.

November (Fig. 3b). Compared to the two other cases, the area affected by extreme wind is more than 3 times larger (Table 1).

Figure 4 shows the synoptic situation during cyclogenesis and the mature stage of Storm Denise. A pronounced trough over France and Germany extends towards the western Mediterranean Sea where it initiates lee cyclogenesis south of the Alps at 00 UTC on 22 November. At this time, both extreme $P$ and $G_{10}$ occur near the cyclone center and behind the cold front, which can be seen in maps of $\theta$ at 850 hPa (Fig. 4b). Twelve hours later, when the cyclone reaches its minimum SLP, the upper-level trough is developing into a cyclonic PV streamer with maximum PV values south of the cyclone center (Fig. 4c). While extreme $P$ still occurs close to the cyclone center and the cold front, extreme $G_{10}$ extends over a large area mainly south of the cyclone center, reaching parts of the North African coast (Fig. 4d).

### 3.2 Case study 2: Storm Jan – January 2023

In January 2023, genesis of Storm Jan happened over the North Atlantic at 18 UTC on 18 January. On 20 January it passed across the Pyrenees towards the Mediterranean, where it intensified by 10 hPa before reaching its maximum intensity of 991 hPa ahead of the Italian coast near Naples at 18 UTC on the same day. On the following day, Storm Jan experienced a small re-intensification over the Adriatic Sea and was categorized as a medicane by EUMETSAT (Eumetsat, 2023), before undergoing cyclolysis at 00 UTC on 23 January. After causing high winds in France and Spain, the storm affected particularly Italy and the Balkans with strong wind gusts and heavy snowfall (AEMET, 2023; European Severe Weather Base, 2023).

Only when the storm reached the European continent, extreme objects were diagnosed as shown in Fig. 5. Reaching the Mediterranean, extreme precipitation affected Sardinia and central Italy (Fig. 5a), while extreme surface winds occurred mainly south of the cyclone center over the sea (Fig. 5b). Figure 6 shows two time steps when large areas were affected by extreme $P$ and $G_{10}$, occurring close to the mature stage of the cyclone. The surface cyclone is positioned below a pronounced large-scale



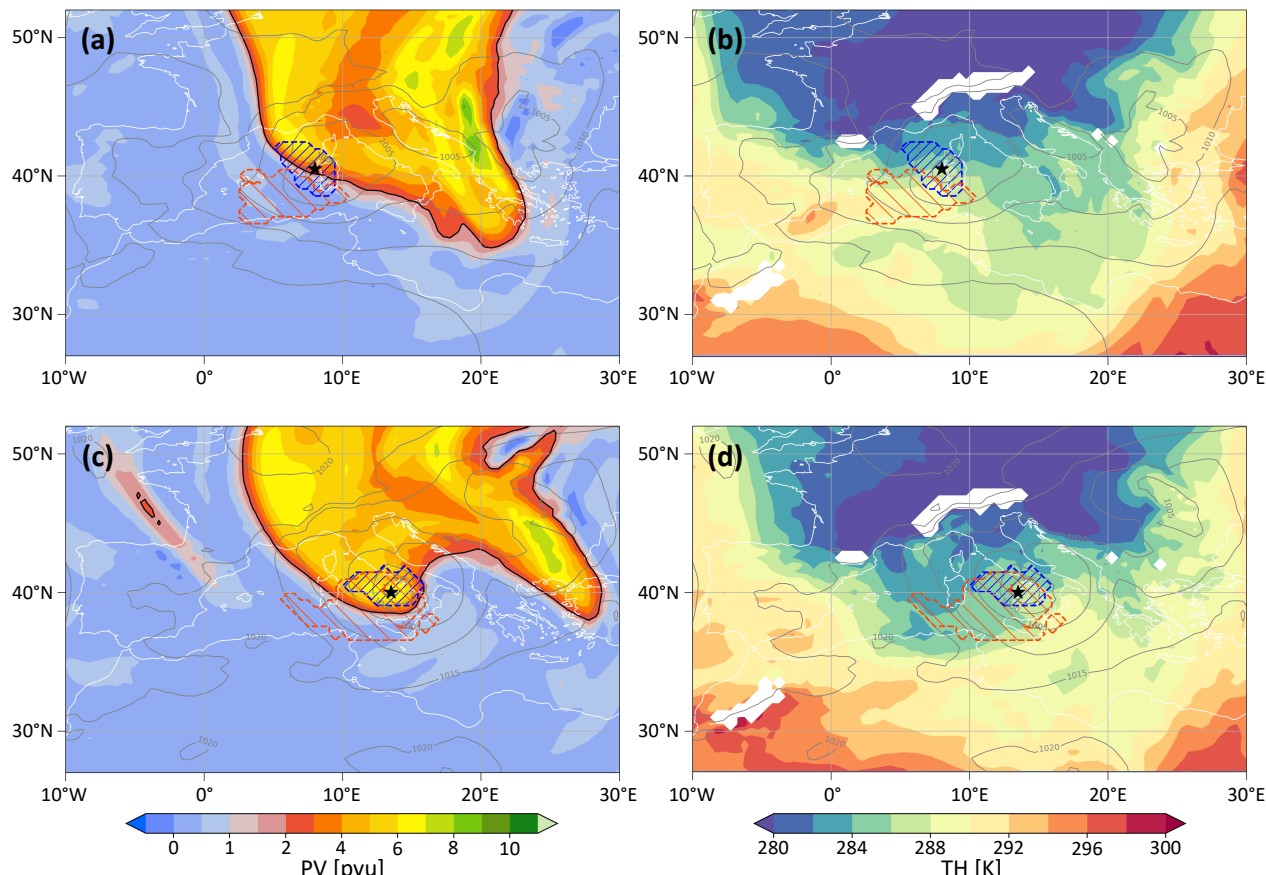

**Figure 6.** Same as Fig. 4 but for Storm Jan at **(a, b)** 06 UTC on 20 January (36 h after genesis) and **(c, d)** 18 UTC on 20 January (time of minimum SLP, 48 h after genesis). PV maps show PV at 310 K.

trough that stretches from Spain towards Greece (Fig. 6a, c). At both time steps, extreme $P$ occurs near the cyclone center and an area of extreme $G_{10}$ extends to the southwest. While extreme winds overlap with an area of warm air at 06 UTC on 20 January (Fig. 6b), they occur mainly behind the cold front 12 h later (Fig. 6d).

### 3.3 Case study 3: Storm Daniel – September 2023

220  At the beginning of September 2023, Storm Daniel hit the headlines as it caused extreme rainfall and devastating floods in parts of Greece and, a week later, northern Libya. Storm Daniel formed at 12 UTC on 4 September off the west coast of Greece. After causing severe flooding in Greece following a first intensification stage on 5 September (AP News, 2023; CBS News, 2023; Greek Reporter, 2023), the cyclone became stationary in the Ionian Sea between Greece and the North African coast. On 10 September, Storm Daniel re-intensified to about 995 hPa and reached the Libyan coast, causing large damage following



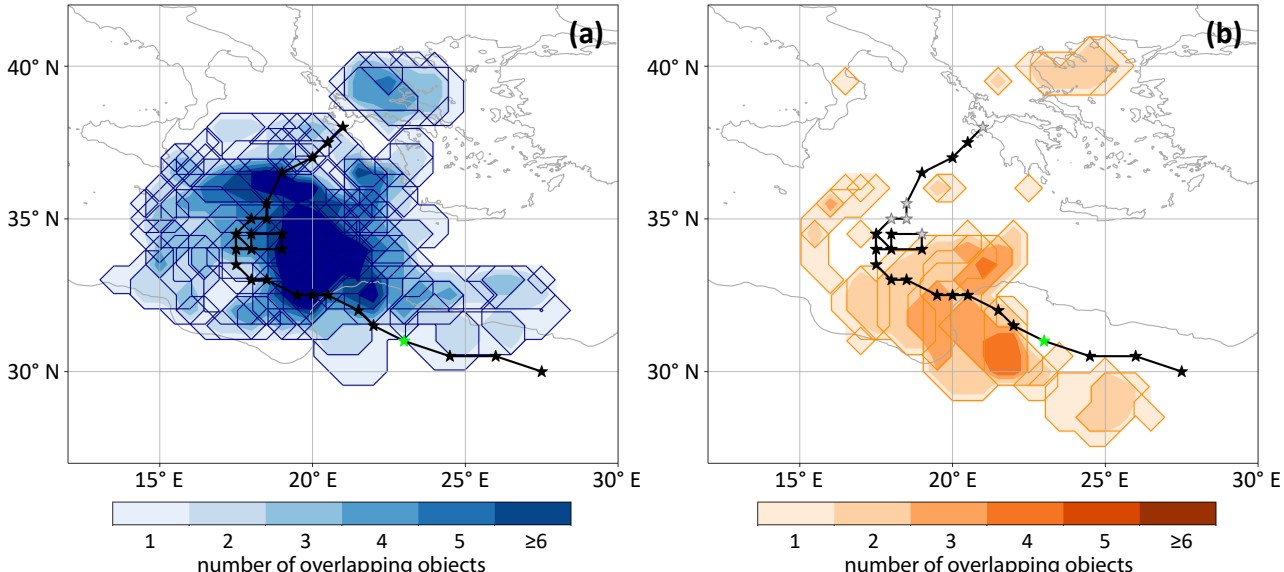

**Figure 7.** Same as Fig. 3 but for Storm Daniel.

the destruction of two dams which led to the flooding of the port city of Derna (Armon et al., 2025). Over 4'000 people were
killed and more than 10'000 people were reported missing (Encyclopaedia Britannica, 2023; NBC News, 2023; WMO, 2023;
reliefweb, 2024).

Figure 7 shows extreme objects related to Storm Daniel. While extreme $P$ occurs in all stages of the cyclone (Fig. 7a),
extreme $G_{10}$ is mainly attributed to the second stage of the storm and most pronounced during its re-intensification on 10
September (Fig. 7b). Due to the stationarity and longevity of the cyclone, parts of the region between Greece and Libya were
affected by extreme rain on almost 3 d and the total area of extreme rain was about four times as large as in the two other cases
(Table 1).

As shown in Fig. 8, Storm Daniel is initially related to a PV streamer stretching from the Ukraine to Italy (Fig. 8a). As
discussed in Flaounas et al. (2024), this streamer resulted from an anticyclonic Rossby wave breaking on the eastern flank of
an omega-blocking pattern over central Europe. The position and orientation of the PV streamer is not unlike the one in the
early phase of Medicane Zorbas in September 2018 (Portmann et al., 2020, their Fig. 2a). During the following days, the PV
streamer breaks up and a PV cutoff remains relatively stationary over the sea, which is likely linked to a stationary blocking
upstream over Central Europe and in agreement with the slow propagation of the surface cyclone until 10 September (Fig. 8c).
The $\theta$-field at 850 hPa (Fig. 8b, d) shows fairly homogeneous values in the region of the cyclone and very high values over
northern Africa. During the first intensification stage, $P$ extremes occur northeast and south of the cyclone center at the edge of





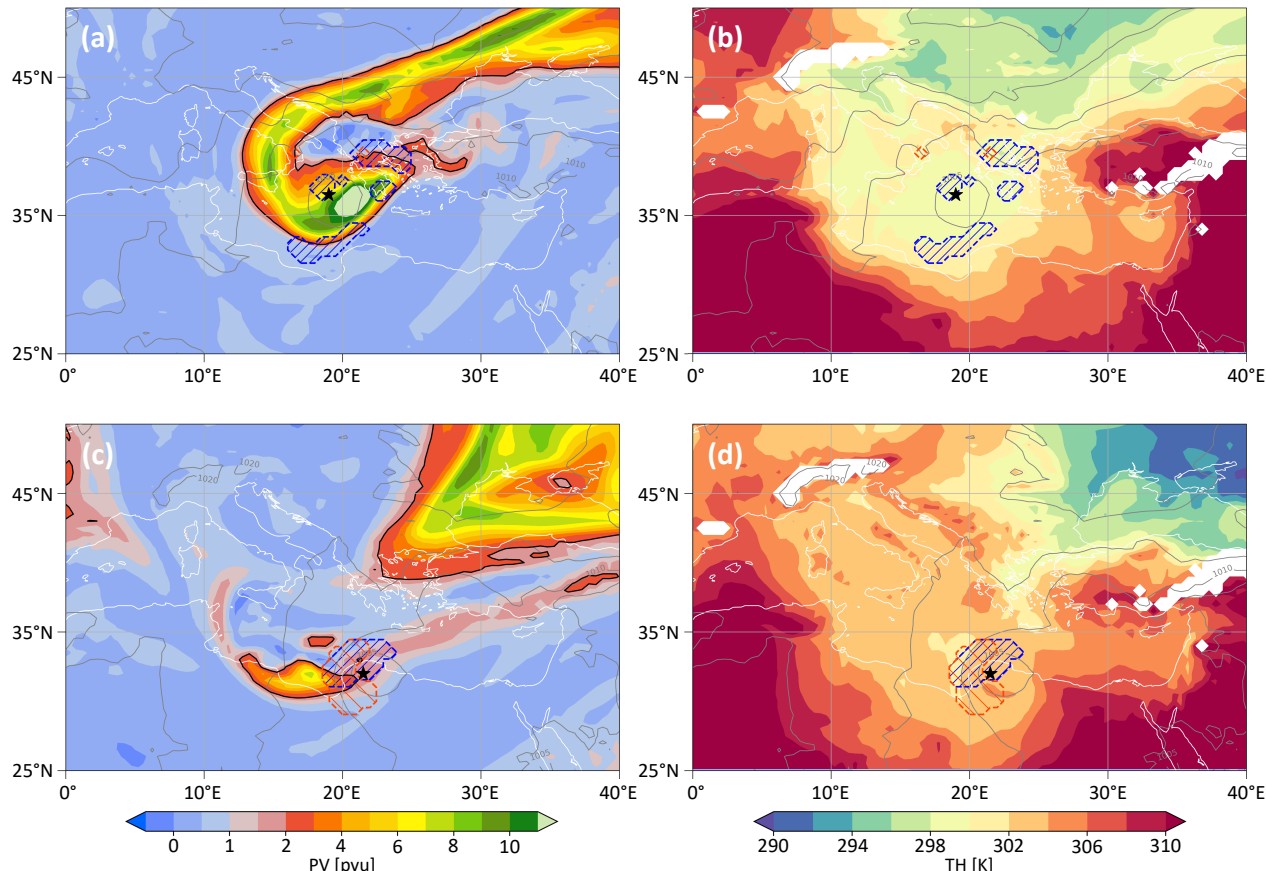

**Figure 8.** Same as Fig. 4 but for Storm Daniel at **(a, b)** 12 UTC on 5 September (24 h after genesis) and **(c, d)** 12 UTC on 10 September (time of minimum SLP, 144 h after genesis). PV maps show PV at 335 K.

the PV streamer (Fig. 8a,b), while extreme $G_{10}$ appears only at single grid points. During re-intensification on 10 September, an area of extreme $P$ is shown northwest of the cyclone center, this time accompanied by extreme $G_{10}$ west of the cyclone

245 center with both objects overlapping with the upper-level PV cutoff (Fig. 8c).

## 4  ECMWF forecast performance

In the following, we assess the performance of ENS in forecasting the cyclone tracks (Sect. 4.1) and the attributed objects of extreme surface weather (Sect. 4.2) depending on the forecast lead time for the three case studies.





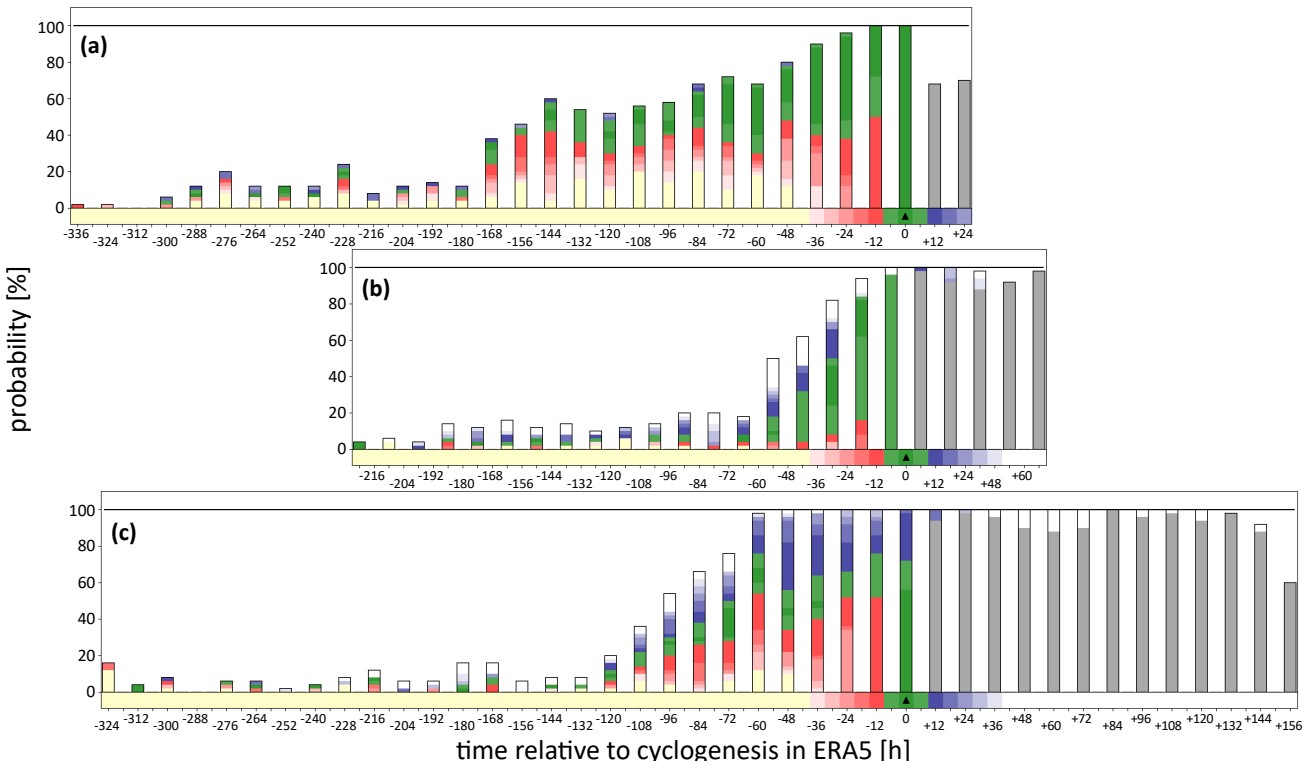

**Figure 9.** Probability of cyclogenesis (along y-axis; in %) in ENS for Storms (a) Denise, (b) Jan, and (c) Daniel for different lead times (along x-axis; in h). Time zero is the time of cyclogenesis in ERA5 (black triangle). Color shading refers to the timing of cyclogenesis in the ENS members (see horizontal bar at the bottom of each panel). Yellow and red colors indicates too early cyclogenesis, blue and white colors too late cyclogenesis, and green an (almost) correct forecast of cyclogenesis time. Grey colors denote members that already have the cyclone at forecast initialization time. Black solid lines denote probability of 100%.

## 4.1 Probability of cyclogenesis

250 Figure 9 shows the probability of a matching cyclone track in ENS, as a function of time relative to cyclogenesis in ERA5, for all three case studies. This diagram requires careful explanation. Every bar corresponds to an ENS forecast, and for instance a bar labeled at the bottom with "−120" represents a forecast initialized 5 d prior to the time of cyclogenesis in ERA5. The height of the bar indicates the percentage of ensemble members that predict a matching cyclone and therefore cyclogenesis, while the colors denote the predicted time of cyclogenesis relative to the actual cyclogenesis time in ERA5. Yellow and red

255 colors signalize too early cyclogenesis in the ENS members (red colors by 12 to 36 h, see the horizontal bars at the bottom of each diagram) and similarly blue, grey and white colors signalize too late cyclogenesis. Green colors denote the (almost)




correct forecast of cyclogenesis time (±6 h).

Let's consider an example for the first case in Fig. 9a: for the forecast initialized 120 h prior to the time of cyclogenesis in ERA5, about 50% of the ensemble members produce a matching cyclone track and therefore predict genesis of this cyclone. The bar is colored about half green and half yellow-red, which indicates that half of the members that predict cyclogenesis do this at the correct time (i.e., at day 5 of the forecast, green), whereas the other half produces cyclogenesis too early, in some cases by 2 d or more (yellow). In contrast, the forecast initialized only 12 h prior to genesis in ERA5 has a vertical bar that extends to 100% and the bar is half green and half red. This indicates that in this forecast all members produce the cyclone, half of them at the right time (±6 h, green), and the other half 12 h too early (red).

Already a brief visual comparison of the diagrams for the three cases reveals clear differences in terms of how early most of the ensemble members predicted the cyclone and whether cyclogenesis, if predicted, occurred rather too early or too late. The three cases are discussed in more detail in the following subsections.

### 4.1.1 Storm Denise

Storm Denise first occurs in ENS 14 d prior to its formation (Fig. 9a). While the probability of cyclogenesis varies between ~10-20% for lead times between 14 to 7 d, a clear increase in probability occurs at −168 h with 40% of all members simulating Storm Denise. While around 60% of all members predict cyclogenesis for lead times between 144 and 96 h, this probability increases further towards shorter lead times, with about half of the members starting to (almost) correctly forecast the cyclogenesis time (green colors). In the forecast initialized 12 h prior to cyclogenesis, all members predict Storm Denise. Interestingly, only 70% of the members include the storm in the forecasts initialized at +12 h and +24 h, when Storm Denise already exists in the ERA5 dataset. A likely reason for this is the fact that Storm Denise was short-lived (42 h, see Table 1) and that at least 3 time steps are required to fulfill the spatial proximity criterion (Sect. 2.3). Therefore tracks in the ensemble members that are (slightly) too short, may not fulfill this condition.

### 4.1.2 Storm Jan

Storm Jan first appears in the forecast about 9 d prior to the time of cyclogenesis in ERA5, and thus had a shorter forecast horizon than Storm Denise (Fig. 9b). However, the probability stays below or around 20% until 66 h prior to cyclogenesis. Only at a lead time of 54 h, an increase in probability up to almost 50% occurs. In the later forecasts, the probability increases steadily until reaching 100% 6 h prior to cyclogenesis as well as 6 and 18 h post cyclogenesis. It is noticeable, that at 54, 42 and 30 h prior to cyclogenesis several members predict the onset of the cyclone too late (blue and white colors), and this also holds for the few members that captured the cyclone at longer lead times. As soon as the cyclone exists in ERA5, the probability remains high (between 90-100%).





### 4.1.3 Storm Daniel

Storm Daniel occurs in ENS for the first time 13.5 d prior to its genesis; however, until 5 d prior to the time of genesis, less than

20% of all members forecast the storm (Fig. 9c). A steady increase in probability to 100% is shown between 120 to 60 h prior to cyclogenesis. After cyclogenesis in ERA5, the forecast probability stays close to 100 % until about 132 h post cyclogenesis, enabled by the longevity of Storm Daniel (see Table 1). Again, it is remarkable that most members miss the actual genesis time of the storm with almost 50% of all members predicting its genesis too early (red colors at $-24$ h and $-12$ h) and about 30-40 % too late (blue colors).

## 4.2 Probability of attributed extreme surface weather

As a next and final step, we assess the performance of ENS in predicting the occurrence of extreme surface weather related to the three Mediterranean cyclones. To this end, objects of extreme $P$ and $G_{10}$ are calculated for each ensemble member as in ERA5 (see Sect. 2.4). Figures 10, 12 and 13 show cyclone-centered probabilities of extreme $P$ and $G_{10}$, as introduced conceptually in Sect. 2.6 and Fig. 2, for the three case studies for different lead times and at different times of the cyclone

lifecycle[2]. From here on, these spatial probability fields are referred to as local probability $p_{loc}$. Again, these figures are rather involved and require some general introduction. They contain many small panels, each for a specific time of the cyclone lifecycle, $t_{cyc}$ (cyclogenesis corresponds to $t_{cyc} = 0$ h), and a specific lead time, $t_{fc}$. The panels are arranged in arrays such that forecast lead time decreases from left to right, and time along the cyclone lifecycle increases from bottom to top. As a consequence of this arrangement, panels that belong to the same ENS forecast occur along diagonals (indicated by red dashed

lines). Given the long lifetime of some of the cyclones and the many possible forecast lead times, only a selection of panels is shown in the figures. The main aim of these arrays of panels is to illustrate how the probability of correctly predicting extreme objects attributed to the cyclones varies with lead time, during different stages of the cyclones' lifecycle, and between extremes of precipitation and surface wind gusts. Additionally, Fig. 11 shows the averaged extreme object probability of ENS within the ERA5 object ($p_{obj}$) at different times along the cyclone lifecycle, to further condense the information shown in the composite

figures of $p_{loc}$.

### 4.2.1 Storm Denise

For Storm Denise, objects of extreme $P$ and $G_{10}$ have been predicted since the first appearance of the storm in the forecast 336 h prior to the time of cyclogenesis in ERA5 (Fig. 9a). Note, that probability values at the beginning of this time range might still occur within the climatological range. In Fig. 10 we focus on lead times of 5 d and shorter, and the times of genesis

($t_{cyc} = 0$ h), minimum SLP ($t_{cyc} = 12$ h), and a time during the decay of the cyclone ($t_{cyc} = 24$ h). As expected, there is an overall increase in the probability of extreme objects with decreasing forecast lead time, however, with interesting differences at distinct time steps along the cyclone lifecycle (Fig. 10). Relatively large objects of extreme $P$ at $t_{cyc} = 0$ h are well forecasted, reaching a $p_{obj}$ of over 70% at short lead times (see also black lines in Fig. 11a). At the time of minimum SLP ($t_{cyc} = 12$ h),

---

[2]Conditional probability maps (see discussion in Sect. 2.4) are shown in the supplement (Figs. S1-S3).





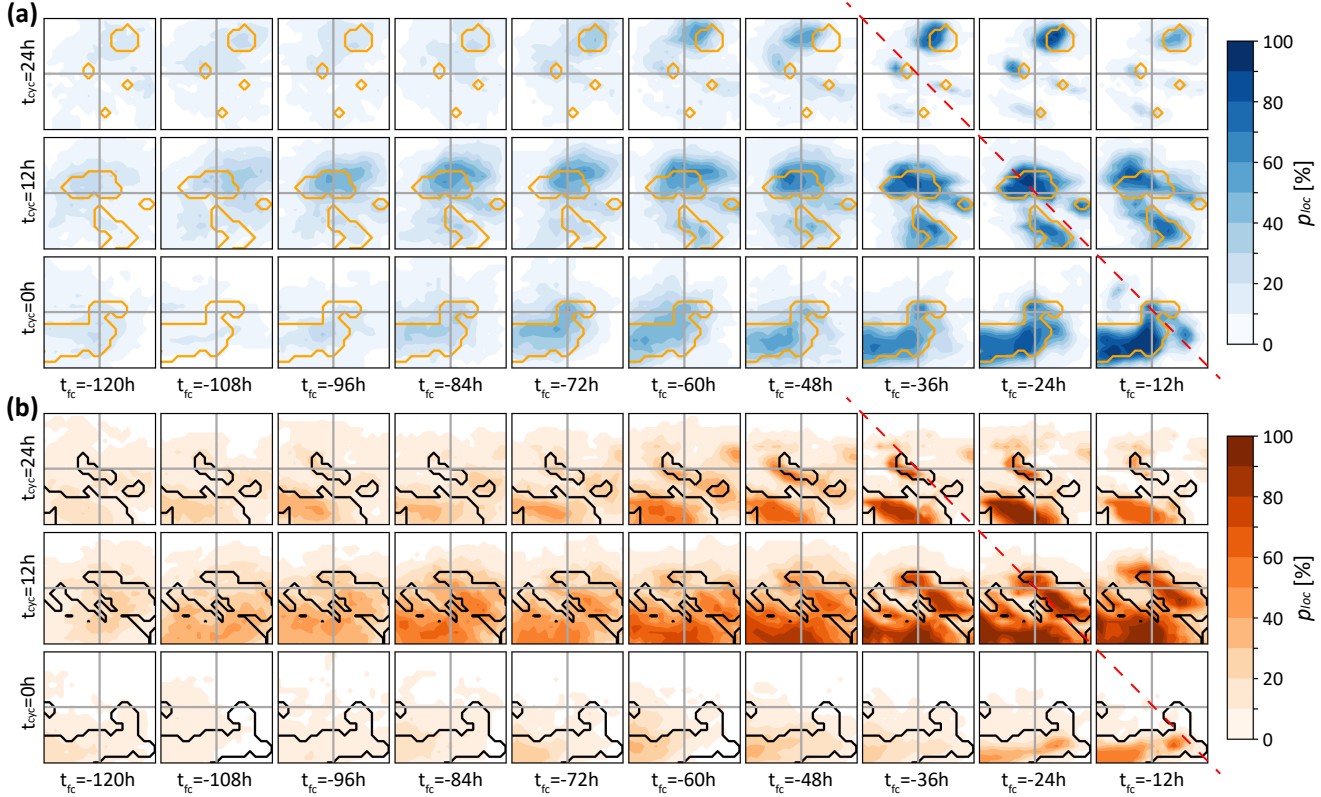

**Figure 10.** Cyclone-centered probabilities ($p_{loc}$) of extreme objects of **(a)** $P$ (blue shading) and **(b)** $G_{10}$ (orange shading) during Storm Denise in ENS (in %). The orange line in (a) and black line in (b) denote the extreme $P$ and $G_{10}$ objects in ERA5, respectively. Lead time decreases along the x-axis ($t_{fc}$; panels every 12 h) and time along the ERA5 cyclone lifecycle increases along the y-axis ($t_{cyc}$; with cyclogenesis at $t_{cyc} = 0$ h). Composites from the same ENS forecast are positioned diagonally as illustrated by the red dashed line.

both $p_{loc}$ and $p_{obj}$ increase already for longer lead times (see also dark grey line in Fig. 11a). For later time steps of the storm, a

relatively large positional error is shown for the comparatively small $P$ objects, leading to a smaller value of $p_{obj}$, which does not exceed 50% (see also light grey lines in Fig. 11a). Overall, it is notable that for lead times of 5 d or longer some members predict the occurrence of extreme $P$ objects, but their position hardly matches the correct location relative to the cyclone center (see relatively uniform low values of $p_{loc}$ in the left panels of Fig. 10a).

For wind extremes, a slightly more steady increase in averaged forecast probability is shown in Fig. 11b, whereby again the probability at $t_{cyc} = 12$ h (maximum intensity; solid dark grey line, see also Fig. 10b) is comparatively high at earlier lead times compared to other time steps within the cyclones' life cycle. Similar to precipitation objects, time steps with smaller objects show reduced probabilities, i.e., more uncertain predictions, revealing difficulties of ENS in simulating such small extreme objects at the right place and time.





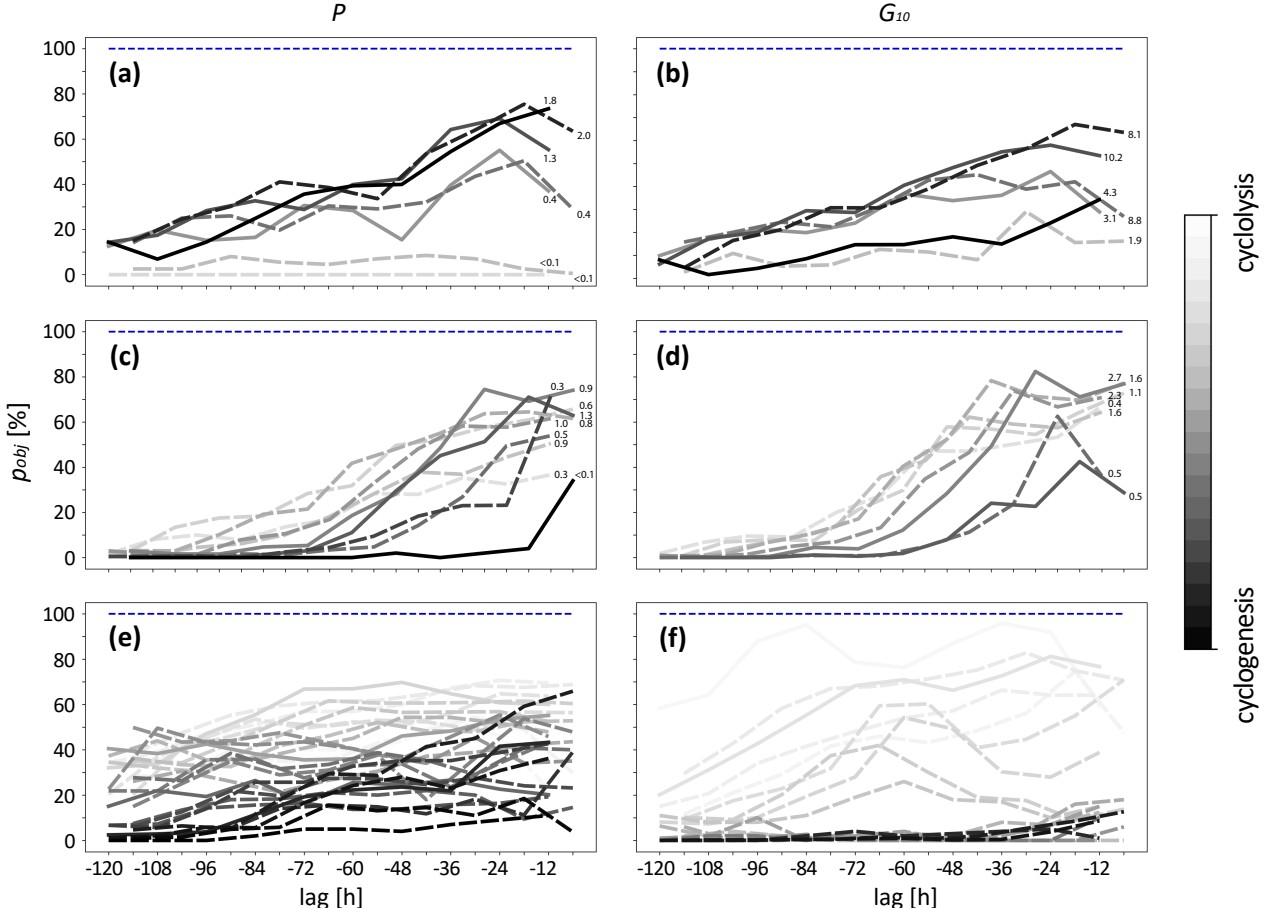

**Figure 11.** Average probability ($p_{obj}$) of extreme objects of **(a, c, e)** $P$ and **(b, d, f)** $G_{10}$ within ERA5 objects (blue and orange contours, respectively, in Figs. 10, 12, and 13) for storms (a, b) Denise, (c, d) Jan and (e, f) Daniel (in %). Each line denotes a separate time along the cyclone lifecycle from cyclogenesis (black line) to cyclolysis (light grey line). Solid lines represent time steps shown in Figs. 10, 12, and 13, other time steps are shown as dashed lines. The area size of the ERA5 object is indicated for each time step as a multiple of $10^5$ km$^2$ next to the lines. Note that here cyclone time steps are shown every 6 h as opposed to every 24 or 36 h in Figs. 10, 12, and 13.

### 4.2.2 Storm Jan

As opposed to Storm Denise, objects of extreme $P$ and $G_{10}$ during Storm Jan are poorly predicted for lead times larger than 48 h, particularly for the early stages of the cyclone life cycle (Figs. 11c,d and 12). Distinct jumps in forecast performance occur at lead times between 48 h and 24 h for the first stage of the cyclone for both $P$ and $G_{10}$ objects (black lines in Fig. 11c, d, $t_{cyc} = 0$ h and $t_{cyc} = 12$ h in Fig. 12). These jumps occur earlier for the later cyclone stage with a significant increase in object probability for lead times between 60 h and 36 h, reaching values of $p_{obj}$ at or slightly below 80% at a lead time of about 48 h







**Figure 12.** Similar to Fig. 10 but for Storm Jan. Note that different time steps along the cyclone life cycle are shown for objects of $P$ and $G_{10}$.

(grey lines in Fig. 11c,d, $t_{cyc} = 24\,\text{h}$ and $t_{cyc} = 36\,\text{h}$ in Fig. 12). In comparison with Storm Denise, the positional error of the objects is smaller and areas with a high $p_{loc}$ match well with the ERA5 objects.

### 4.2.3 Storm Daniel

Figure 13 shows cyclone-centered object probabilities for Storm Daniel. Incoherent objects of extreme $P$ in the early stage of the cyclone are poorly represented in ENS (Fig. 13a, $t_{cyc} = 24\,\text{h}$ and $t_{cyc} = 60\,\text{h}$; black lines in Fig. 11e). Thereby it is important to consider that in case of $t_{cyc} = 24\,\text{h}$, only 20% of all members contain the cyclone for a lead time of 96 h and still only





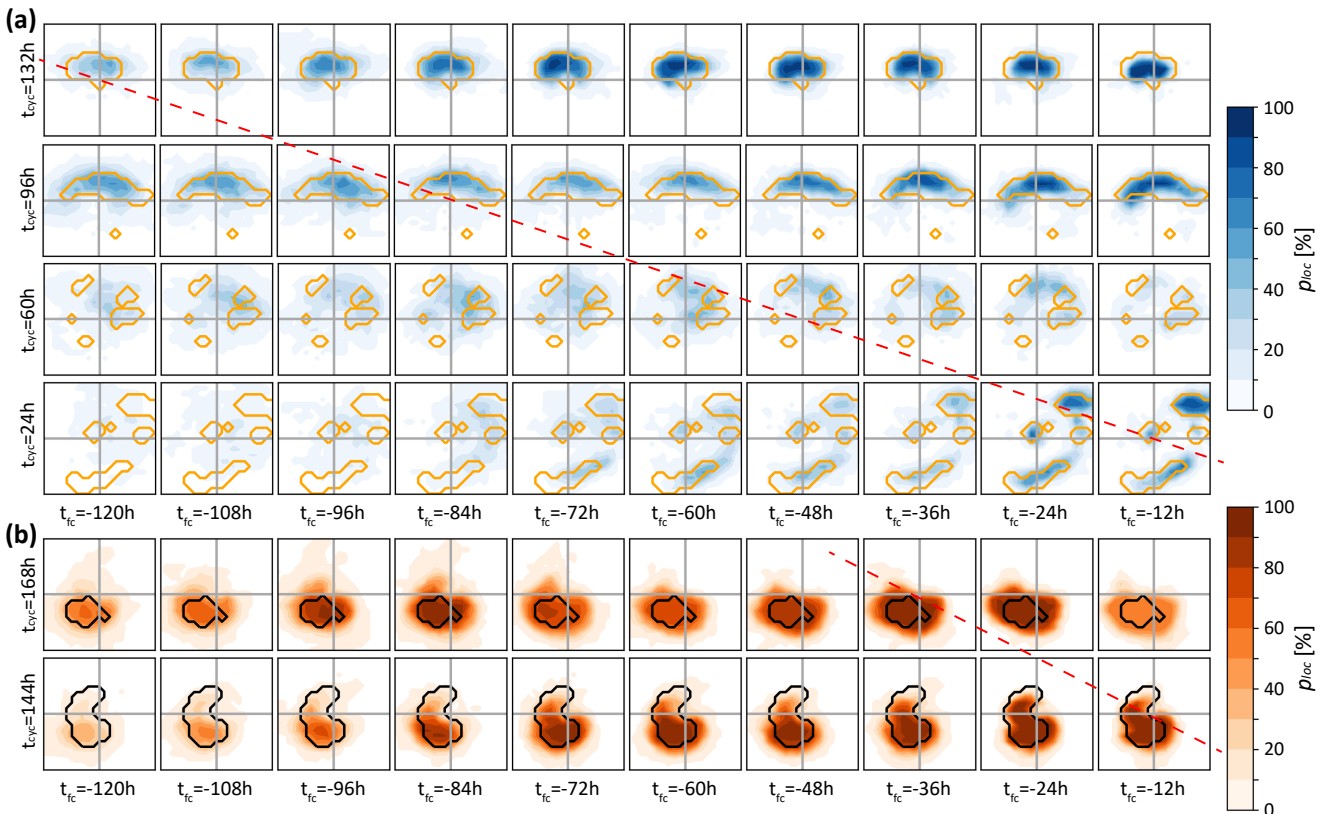

**Figure 13.** Similar to Fig. 10 but for Storm Daniel. Note that different time steps along the cyclone life cycle are shown for objects of $P$ and $G_{10}$.

50% of all members for a lead time of 72 h (see Fig. 9c). As extreme $P$ objects become more coherent during the later phase of the cyclone lifecycle, for example, at $t_{cyc} = 96$ h and $t_{cyc} = 132$ h, and the storm is already existing in ENS for shorter lead times, $p_{obj}$ reaches values up to 40% for lead times up to 5 d. The shape of these objects is represented well by the forecast
with over 80% of the members capturing large parts of the object in ERA5, e.g., for a lead time of 72 h in case of $t_{cyc} = 132$ h.

The extreme $G_{10}$ objects that occur during the late stage of Storm Daniel are predicted well in advance, reaching a $p_{obj}$ of around 60% for a lead time of 4 d (light grey lines in Fig. 11f) with values of $p_{loc}$ exceeding 80% (Fig. 13b). It is noticeable that the ensemble slightly overestimates the size of these objects, particularly for $t_{cyc} = 168$ h. Very small objects of extreme
wind gusts in the early stage of the cyclone that only cover few grid points are poorly or not represented in ENS (not shown).



## 5 Discussion and conclusions

In this study we assess the performance of operational IFS ensemble forecasts (ENS) in predicting the occurrence of Mediterranean cyclones and attributed objects of extreme precipitation ($P$) and surface wind gusts ($G_{10}$). For this aim we introduce a method that is based on several pragmatic choices and can be applied quasi-operationally. We argue that such quasi-operational "online" (or "on-the-fly") object-based methods will become increasingly important in the future, because the rapidly increasing size of ensemble forecast data will make retrospective "offline" analyses of this kind much more difficult and cost-intensive. Our method includes the following steps:

1. Identify matching cyclones between ERA5 and every ENS member based on spatio-temporal criteria.

2. Define two-dimensional objects of extreme surface weather in both ERA5 and ENS members using ERA5 percentiles as thresholds.

3. Attribute cyclones to extreme weather objects, separately in ERA5 and every ENS member, based on overlap criteria.

4. Calculate ensemble mean cyclone-relative probabilities of predicting extreme objects of $P$ and $G_{10}$ for different forecast lead times and times along the cyclone life cycle.

To illustrate and test this method, we selected three Mediterranean cyclones with different characteristics. The first case, Storm Denise, has a short lifetime of 42 h and occurs on the front side of a pronounced upper-level PV streamer over the Gulf of Genoa. It is the most intense of three three storms in terms of minimum central SLP. The second case, Storm Jan, originates in the North Atlantic and propagates into the Mediterranean along the rearward flank of an upper-level trough. Storm Jan is the fastest moving and intensifying of the three cyclones. Finally, the third case, Storm Daniel, has an exceptionally long lifetime of 174 h. It is associated with a quasi-stationary upper-level PV streamer and later PV cutoff. Storm Daniel caused persistent extreme $P$ in large areas leading to flooding in both Greece and Libya, while Storm Denise had the largest footprint of extreme surface winds. For these three cases, extreme $P$ usually occurs around the storm center, while extreme $G_{10}$ is mostly located south of the storm center.

The analysis of the three cyclones in the ENS dataset shows a general increase in probability of both the occurrence of the cyclone track itself and of the attributed extreme surface weather with decreasing forecast lead time. In particular, we find a high forecast probability for extreme objects for lead times $\leq 48$ h. For longer lead times, our study reveals a large case-to-case variability in the predictability of both the cyclone track and its attributed extreme surface weather. While 40% of the ensemble members predict Storm Denise 7 d prior to its genesis, the same value is reached only 4.5 d prior to genesis of Storm Daniel and only 2.5 d prior to genesis of Storm Jan. These results are consistent with Doiteau et al. (2024) who found that on average more intense storms in terms of minimum SLP (such as Storm Denise) are detected by more ensemble members at longer lead times, and that rapidly-intensifying storms (such as Storm Jan) show a particularly low forecast skill. In the case of Storm Denise, an initial increase in forecast probability of the cyclone track is followed by a stagnant phase between 7 and 4 d prior



to cyclogenesis and a gradual increase afterwards. In contrast, Storms Jan and Daniel exhibit a more rapid increase in forecast probability of their tracks. The actual formation of the storms is predicted on average slightly too early for Storms Denise and Daniel, and slightly too late for Storm Jan. Late forecasts initialized after cyclogenesis in ERA5 show a probability of 90–100% for Storms Jan and Daniel, matching findings by Di Muzio et al. (2019) about the higher accuracy of later forecasts that are initialized after cyclogenesis. For Storm Denise, ENS shows a notable reduction in probability to less than 70 % at this stage, which is probably linked to the short lifetime of this storm, which could cause less members fulfilling the spatio-temporal criterion for matching cyclone tracks between ENS and ERA5.

The predictability of objects of extreme surface weather exhibits a similarly strong case-to-case variability. For Storm Denise, such objects are already predicted in up to 40% of all members at a lead time of 120 h, followed by a staggered probability increase at lead times of 72 h and 24 h. However, despite the early detection of the objects, a relatively large positional error remains also for short lead times, particularly for extreme $P$ objects. Distinct jumps in object probabilities occur for the forecast of extreme surface weather attributed to Storms Jan and Daniel, which both feature comparatively small positional errors of such objects relative to the storm center.

Although three cases are not enough to draw robust generalized conclusions, our case studies indicate three key aspects that affect the probability of extreme surface weather objects in ENS forecasts:

1. *Object size:* We find that larger and more coherent objects are usually better represented compared to multiple small objects. This partially follows by design, as we define such objects with a (percentile-based) threshold. For small objects, values of $P$ and $G_{10}$ are usually close to the threshold value, resulting in a higher ensemble uncertainty of simulating the object.

2. *Cyclone track:* A better representation of the cyclone in ENS results in an improved predictability of extreme surface weather objects at longer lead times. In case of Storm Denise, which is already simulated by ∼50% of the ensemble members a week prior to its actual formation, the forecast of extreme surface weather objects is less uncertain compared to Storms Jan and Daniel, which are captured by fewer ensemble members at similar lead times. This is illustrated by Fig. S4 in the Supplement, which shows conditional probabilities of surface extreme weather objects as opposed to the unconditional probabilities shown in Fig. 11. Compared to the other two cases, conditional probabilities are higher for Storm Denise at long lead times, indicating that for this cyclone, members including the storm already have a good representation of the attributed extreme weather compared to members including Storms Jan and Daniel at the same lead times.

3. *Storm lifecycle:* We reveal an increase in the probability of predicting extreme weather objects with increasing cyclone lifetime for similar lead times. This likely is a consequence of the existence of the cyclone in almost all members in forecasts of later storm stages, while forecasts for early storm stages are affected by a significant amount of members



not including the storm. In essence, this indicates that for the cases investigated here, the process of cyclogenesis is more uncertain, i.e., more challenging to simulate, than the later cyclone intensification and decay.

Overall, we show that, for the three cyclones investigated, ENS predicts objects of extreme surface weather very well with respect to the storm center for lead times up to about 2 d. For longer lead times, the forecast uncertainty of extremes appears to be strongly case dependent. We conclude that the proposed methods can yield meaningful information about the ensemble prediction of surface weather extremes and we plan to apply the method systematically to a multi-year dataset of Mediterranean cyclone forecasts. However, it is also important to mention caveats of this study: (i) we analyze the probability of objects only relative to the cyclone center without investigating the error in the cyclone center position; (ii) we only quantify whether ensemble members exceed a certain threshold, but not by how much (which would be relevant, e.g., for flood prediction in case of extreme $P$ objects); (iii) to facilitate the comparison we compared ERA5 and ENS on a grid with 0.5° horizontal grid spacing and reduced the temporal resolution of the ERA5 cyclone tracks to 6 h, which impedes a more detailed analysis and taking full advantage of the respective dataset resolution.

As mentioned in the second part of this study, we plan to apply the method introduced here to a large set of Mediterranean cyclones. This serves to potentially identify differences in forecast performance between cyclones with different characteristics, for instance in terms of their upper-level PV signature at time of maximum intensity (Givon et al., 2024) and of upstream processes over the North Atlantic that might influence the dynamical evolution and forecasts performance over the Mediterranean (Raveh-Rubin and Flaounas, 2017; Portmann et al., 2020; Scherrmann et al., 2024). Such analysis will improve our understanding of the large case-to-case variability found in this study, and potentially of the underlying causes for forecast errors in terms of intensity, location and area size of surface weather extremes.

*Code and data availability.* The ERA5 dataset can be downloaded from the Copernicus Climate Data Store https://climate.copernicus.eu/climate-reanalysis; (https://climate.copernicus.eu/climate-reanalysis; Copernicus Climate Service, 2025). The ENS surface fields of SLP, $P$, and $G_{10}$ will be made available at the ETH research collection at the time of acceptance of this paper. Scripts used to produce the analyses and figures in this study are available on request from the authors.

*Author contributions.* KH performed the analyses, produced all figures and wrote the initial draft of the manuscript. DB developed the quasi-operational retrieval, cyclone identification, and storage procedure of ENS data from the ECMWF to ETH Zurich. KH and HW designed the study and discussed the results. All authors helped to improve the manuscript.

*Competing interests.* Some authors are members of the editorial board of *Weather and Climate Dynamics*. The authors declare that they have no other competing interests.



*Acknowledgements.* This research has been supported by the Schweizerischer Nationalfonds zur Förderung der Wissenschaftlichen Forschung (grant no. 205419). We are most grateful to Urs Beyerle and Michael Sprenger for their help with the data handling and automatization of the method, to Manos Flaounas and Shira Raveh-Rubin for discussions and feedback on earlier versions of the manuscript, and to Jacopo Riboldi, Franziska Schnyder, Iris Thurnherr, and Marc Federer for discussions about the design of the figures.



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
