# Peer review of "Quantifying forecast uncertainty of Mediterranean cyclone-related surface weather extremes in ECMWF ensemble forecasts. Part 1: Method and case studies"

_EGUsphere, 2025_

## Referee Comment (RC1)

**Article review**
**"Quantifying forecast uncertainty of Mediterranean cyclone-related surface weather extremes in ECMWF ensemble forecasts.**
**Part 1: Method and case studies"**

**Summary:** In their article, [Katharina Hartmuth et al.] present a novel method to assess the forecast skill of the ECMWF ensemble in predicting extreme weather events associated with Mediterranean cyclones. The first part of their study explains the methodology and illustrates it with three case studies of impactful Mediterranean cyclones. The forecast skill is evaluated based on the ability to predict the occurrence of extreme precipitation and extreme surface winds (both defined by exceedance of the local 99th percentile).

The paper is overall clear and well-structured. Particular attention has been given to the description of figures, which makes them especially pleasant to read. In the following, you will find some suggestions, mostly minor, that may help improve the paper.

**1 Major revisions**

**0.5°, 6 h:** As discussed in your conclusion, using 0.5° may be limiting, especially if you look at small objects (such as medicanes). Also, 6 h is coarse for the Mediterranean, where storms evolve quickly. If the work is not too big, I strongly encourage you to take full advantage of the available resolutions. Another way (if increasing resolutions is not possible) could be to use products like "accumulation of precipitation within the 6 h" or "maximum wind gust within the 6 h" if they are available.

**Radius of 400 km:** While this radius is adapted for medicanes, it is not for strong winds associated with extratropical cyclones, even in the Mediterranean. I suggest using a circle of 1000 km around the cyclone centre, or (maybe better) using isolines of pressure that you utilised already to capture cyclones. Also, if you want to avoid attributing high wind to weak lows, you could use a threshold on the minimum central pressure.

**Percentile calculated each season:** I do not think that this is relevant when looking at impacts. Indeed, high wind or precipitation do not impact differently following the season but following their strength. I encourage you to recalculate the results based on a fixed threshold for the whole year. Also, generally the 98th percentile has been used for wind gusts [Klawa and Ulbrich (2003)], as it was shown to fit well the observed losses. Finally, you could use the so-called Storm Severity Index to draw conclusions on the prediction of the impacts.

**"Probablity":** The word probability is sometimes unclear throughout the manuscript. You could reword it in the text when it is relevant, like "proportion of members". The terms "conditional" and "unconditional" probabilities are also adding complexity that may be avoided. This point is a detail, but I think the manuscript could gain clarity with slight modifications.

**Different operational cycles of IFS:** You plan to use different cycles of the IFS (successive improvements of the model) for your statistical analysis. I suggest either using reforecasts instead, or clearly justifying how the use of different cycles will—or will not—affect your results.

**2 Minor suggestions**

**Title:** The title could be more concise and clearer, maybe something like: Forecast uncertainty of high-impact weather associated with Mediterranean cyclones. Part 1: Method and case studies. Also, the term "forecast uncertainty" may be replaced by "predictability" in your case.

**General comment:** Using the passive voice may provide better objectivity.

**General comment:** A figure illustrating the value of the local 99th percentile of precipitation and wind gusts would be very enjoyable.

**Introduction**
**L. 4:** "ECMWF" not defined at this point.

**L. 5:** "We apply .. to attribute Mediterranean cyclones to events of extreme.." Is it not the opposite? (We apply .. to attribute extreme events to Mediterranean cyclones).

**L. 25:** The whole sentence could be split into two parts for clarity ('Jansà .. cyclones', 'surface' .. '2024)'.

**L. 36-37:** How can you say that the forecast skill of the position is higher than for the intensity since it is based on two different parameters?

**L. 59-64:** Could be reworded to shorten the sentences and prevent the repetition of "such".

**L. 66-80:** This part should be made more concise and clearer. For example, the methodology presented in the "Discussion and Conclusions" is easier to follow. The word "whether" line 70 could be replaced, as uncertainty does depend (to a lesser or greater extent) on the following parameters.

**L. 75-77:** "For instance, .. 1000 individual forecasts." is unnecessary.

**L. 75:** ERA5 not defined at this point.

**L. 79:** "This first part .. (1) and parts of (2) and (3)." May be reworded.

**L. 82:** ERA5 not defined at this point.

**Data and method**
**L. 89-90:** Passive voice may be more appropriate here.

**L. 98:** Replace "Reduced sea level pressure (SLP)" by "Mean Sea Level Pressure (SLP)"

**L. 107:** Replace "active" by "operational"

**L. 112:** "data is downloaded every 6 h": Not very clear. This whole sentence could be included at the beginning of the section. "ECMWF ensemble consists of 50 members initialised twice per day. For each base time, a forecast output is available every 6 h up to a maximum lead time of 15 d."

**L. 137:** Can we be sure it is the same cyclone? Does it happen frequently? If yes, maybe a criterion based on the SLP value may be helpful.

$\theta$: Usually, $\theta_e$ or $\theta'_w$ are better adapted to look at fronts.

**L. 158:** "probability of such object". Not very clear, is it a probability of occurrence? If yes, please clarify in the text.

**Fig. 3, 5, 7:** "number of overlapping objects" refers to extreme events detected in X members. Maybe the legend could be reworded to enhance clarity. The stars are too small.

**Case study overview**

**Table 1:** "Maximum intensification" may be in [hPa / 12 h] for the three cases to enhance clarity and to fix the given unit.

**Fig. 4, 6, 8:** I suggest slight modifications to these nice looking figures. White coastlines a bit thicker, grey contours a bit thicker, black star larger, TH [K] to $\theta$ [K]. The areas are also difficult to see with the colours (blue on blue and orange on orange).

**L. 206:** I do not think it "passed across the Pyrenees", at least for the storm centre. It seems that it remains on their North side.

**L. 206:** by 10 hPa / how much time?

**L. 208:** "as a medicane by EUMETSAT": I would be careful with this sentence, since the actual definition given by [Miglietta et al., 2025] is more precise. A consensus did not exist at the time EUMETSAT did that categorisation for a storm evolving in the middle of winter. In particular, "the warm core" was not formally detected. Also, if you choose to keep the sentence, the reference given should be corrected in L. 482 (northern Atlantic? + double https)

**ECMWF forecast performance**

**L. 246:** I would replace the word "performance" by "skill" in the whole manuscript since we compare to ERA5.

**Fig. 9:** The figure was difficult to read, but the description was really useful. However, I suggest two simplifications: replace the word "probability" by "percentage of members", or plot directly the number of members. Replace "time relative to cyclogenesis in ERA5" by something like "Initialization time [h]". The parts after "0" seem not very useful for the analysis and may be removed.

**L. 301:** "They contain many small panels". Replace by "Each panel represents.."

**L. 307:** "averaged extreme object probability of ENS within the ERA5 object", may be simplified.

**Fig. 10, 12, 13:** The figures could be simplified by plotting $t_{fc}$ at only two times (cyclogenesis and mature stage). It would also have the advantage of avoiding a supplementary notation ($t_{cyc}$) while providing useful information. Also, if you modify Fig. 9. as stated earlier, maybe replace ($t_{fc}$) by "Initialization" or "$t_{ini}$" for consistency.

**Fig. 11:** If you do modify Fig. 10, 12, 13, consider modifying Fig. 11 by plotting only cyclogenesis and the mature stage. If you want to keep the area size, make it larger. It would be easier to understand, and you can still provide additional information in your part 2 if it is meaningful. And last, maybe the figure is misplaced; I would expect this information after reviewing the 3 case studies.

**Conclusion**
**L. 366:** "three three"

**L. 371:** This sentence could be discussed a bit more. Indeed, we expect the strongest winds to be located in the southern parts of extratropical cyclones, but in medicanes they tend to occur all around the centre.

**L. 400:** Could be discussed for medicanes that are small objects with high precipitation.

**L. 435:** It would be also interesting to see how prediction varies depending on cyclone type (baroclinic vs. strongly diabatically impacted).

---

## Community Comment (CC1)

**Review of "Quantifying forecast uncertainty of Mediterranean cyclone-related surface weather extremes in ECMWF ensemble forecasts. Part 1: Method and case studies" by Hartmuth et al.**

Review by Michael Schutte

Recommendation: Accept subject to minor revisions

The study focuses on forecast uncertainty of Mediterranean cyclones with respect to the cyclones' predictability, and their related wind and precipitation extremes. Based on ensemble forecasts from ECMWF, the authors outline a novel methodology to track and assess the representation of cyclone-related weather extremes. Three case studies illustrate the methodology and highlight case-to-case variability in forecast uncertainty of those extremes. Additionally, forecast uncertainty is greater if smaller regions are affected by extremes, during earlier lead times, and if the cyclone is not well captured by the ensembles.

Although my expertise lies not in Mediterranean storms, the methodology appears to yield valuable results. At the same time, the manuscript could be strengthened by considering the following aspects:

- Varying dependence between forecast uncertainty and cyclone lifetime
- A risk for confusing the goals stated in the introduction and those actually addressed by the paper
- Potential differences in bias between ENS and ERA5
- The fairness of comparing predictions of extremes for storms of different lifetimes

These points will be discussed further in the following.

**Main comments**

You note in several places (e.g., l. 11-12 and l. 413-417) that predictions are more uncertain during the earlier stages of the cyclones' lifecycle. While this is consistent with Fig. 11 for storms Jan and Daniel, storm Denise appears to show the opposite behavior, with uncertainty being highest during the later stages of its lifetime. It may therefore be helpful to acknowledge this varying dependence between forecast uncertainty and cyclone lifetime, rather than presenting it as a general tendency.

Paragraph in l. 66-80: As a reader it is not immediately clear, which goals you aim to achieve in this study and which you investigate in the second part. It could be beneficial to highlight that these goals apply to both studies together in l. 67. Alternatively, the authors could also mention the goals relevant to this paper at first and outline goals for the second part separately.

l. 142-146: Considering the different IFS model cycles, how well do these thresholds align between ERA5 and ENS? Is the difference between ENS and ERA5 small enough to be negligible or might it have an effect on how well extreme objects are detected in the

ensembles? It could be good to mention this, as it might limit the interpretability of the results otherwise.

You compare associated extremes and cyclones of different lifetimes. I think the three storms are well chosen, as they also reflect this aspect. However, one might have to be careful when comparing the results of the different storms with each other. For example, in Fig. 9, it is hardly possible to predict a later cyclogenesis for storm Denise, due to its very short lifetime. Thus, the algorithm might miss out any forecast that predicts a storm some days later, opposed to storm Daniel, with potential consequences for the predictability of P and $G_{10}$. The authors could at least point out this aspect, e.g., in Sec. 4.1.1.

**Minor comments**

l. 7-8: The authors could define in the abstract what they mean with 'objects of extreme surface weather' These two-dimensional objects are defined nicely in Sec. 2.4., but it might be helpful to provide a short explanation in the abstract, as well.

l. 12: The authors could add a sentence about implications at the end of the abstract to highlight the relevance of the study's results.

l. 34 'cyclone speed': As a reader it is not immediately clear what is meant with cyclone speed. Do the authors mean the propagation speed of cyclones, intensification speed/cyclogenesis, or 10-m wind speeds?

l. 106-107: Since you mention the region that the retrieved fields cover, you could also add the coordinates of the domain in parentheses.

l. 107: Why did you chose 0.5° grid spacing? As you also mention in the conclusions, a higher spatial resolution could be valuable in the case of investigating Mediterranean cyclones, e.g., with respect to cyclone detection and tracking (Aragão et Porcù, 2022).

Fig. 1: I really like this figure to visualize how the cyclone track matching algorithm works. Unfortunately, you don't refer to it in the text right now. For example, this could be done at the end of Sec. 2.3.

l. 136-138: How much does this methodological choice increase the number of 'random jumps' in identified cyclone tracks, e.g., in Figure 1, jumping from the first match to the second match and then to the third match? And does this have any potential drawbacks for the analysis of forecast uncertainty? For example, a jump in detected cyclone could result in a sudden change of P or $G_{10}$ probabilities due to a different location of cyclone center.

l. 169-179: The first paragraph of Sec. 2.6 elaborates on the probability of extreme objects, in line with its title. However, the second paragraph justifies why the three storms have been selected as case studies. Thus, it could be beneficial to have a separate section for the second paragraph, e.g., 'Selection of case studies', or revising the title of Sec. 2.6.

Tab. 1: Are the SLP minimum values based on ERA5 data or on observations? Furthermore, you have mentioned socioeconomic impacts of Mediterranean cyclones, so I would suggest to add one more column for only land area/grid points affected by P, and in the same way one column for $G_{10}$.

Fig. 9: I really like how you visualize the probability of cyclogenesis in this figure. As especially subpanel (c) is rather long, the readability could be improved if you add thin horizontal lines, e.g., every 20%, in all subpanels instead of having only the 100% line.

l. 259-263: At lead time -120 h in Fig. 9a, you mention that half of the members are predicting cyclogenesis at the correct time and half too early. However, there is also a small portion colored in blue, indicating that a few members predict cyclogenesis too late. You could either mention that in the text, or choose a different lead time, e.g., -60 h.

l. 266: Could these differences be linked to different origins, i.e., the North Atlantic vs. the Mediterranean?

Fig. 10: Do the boxes shown in each small subpanel have the same spatial extent as in Fig. 2, i.e., 20° latitude by 20° longitude?

Fig. 11a: Here and in Fig. 10a, would it be possible to elaborate shortly, why $p_{obj}$ of P decreases for the -12 h lead time? This seems to be consistent for several times along the lifecycle with exception of earliest time at cyclogenesis.

l. 425-427: You could comment on why a higher temporal and spatial resolution might matter for your analysis of Mediterranean cyclones, e.g., an improved detection of precipitation and wind extremes.

**Technical comments**

l. 89: You could use a comma after 'Then', i.e., 'Then, we present…', to improve readability.

l. 237-238: The sentence 'The position and orientation of the PV streamer is not unlike the one in the early phase of Medicane Zorbas in September 2018' is understandable, but it has some grammatical inconsistencies and could be expressed more clearly. Specifically, 'position and orientation" forms a compound subject, so the verb should also be plural.

**References**

Aragão, L., Porcù, F. Cyclonic activity in the Mediterranean region from a high-resolution perspective using ECMWF ERA5 dataset. Clim Dyn 58, 1293–1310 (2022). https://doi.org/10.1007/s00382-021-05963-x

---

## Author Response (AR1)

*Paper egusphere-2025-4111*

**Quantifying forecast uncertainty of Mediterranean cyclone-related surface weather extremes in ECMWF ensemble forecasts. Part 1: Method and case studies**

by Katharina Hartmuth, Dominik Büeler, and Heini Wernli

We are grateful to the three official reviewers and to Michael Schutte for their detailed and constructive comments that certainly help us to further improve the manuscript. Based on the reviewers' suggestions, several changes in the manuscript have been implemented as outlined below. The most important aspects of our replies and revisions have been:

- As suggested by all reviewers, we clarified the framing and main objectives of the study, and particularly the role of part 1 vs. part 2.
- We clarified used terminologies such as "(un-/conditional) probability" and better highlight methodological choices and their limitations.
- We improved some of the figures to enhance their readability despite their complexity.
- To strengthen our synthesis and the comparative discussion of the case studies, we added a summary table in the concluding section.

However, we also would like to explain upfront that maybe the referees underestimated to a certain degree the complexity of the data processing machinery we built over the last years in order to enable this study (and more statistical analyses in a future Part 2 paper). We obtained this impression from the several suggestions about repeating our analysis with higher resolution and more fields etc. – although these suggestions are reasonable and scientifically interesting, they are, unfortunately, not feasible, as we try to explain in this paragraph. Therefore, for reasons of feasibility, we have not been able to pick up all suggestions by the referees (see point-by-point responses below). Here we provide more information about our data processing machinery (and some of this information has been added to the revised manuscript, see for example added paragraph in L123f):

- First, it is the intention of this study to look at operational IFS ensemble forecasts as opposed to re-forecasts/hindcasts. Both approaches are interesting and have their pros and cons (see also below), but we aim at investigating the performance of operational forecasts. The reason is that operational forecasts are run with the highest native resolution and number of ensemble members, and come with the full set of model levels required for some of our analysis we plan for a future Part 2 study, as opposed to their re-forecast/hindcast counterparts (see further details below). To achieve this, we developed a procedure to retrieve the newest forecast data from ECWMF twice daily in real time and save it on the servers of our group, where we run a postprocessing to obtain netCDF files, identify cyclones and warm conveyor belts, and calculate backward trajectories.

- One goal of this time-critical procedure is to preserve model level data, which is only available for a few days at the ECMWF server before being deleted. (In the MARS archive at ECMWF, only ensemble data on a few pressure levels is available. We thus compiled a unique multi-yearly ECMWF forecast dataset that not even ECMWF itself possesses.) While so far, this 3D data has been used mainly by other projects in our group, it is our aim to use it for the calculation of trajectories and upper-level PV in the continuation of this project.
- For the current spatio-temporal resolution (0.5°x0.5°, 6 h), we download about 240 GB of data per day, which adds up to about 7 TB per month. After postprocessing, we end up with about 19 TB of data each month. Collecting data from 2022 with the aim of having a complete dataset for SON and DJF, the total size of the dataset will soon exceed 400 TB. The resolution suggested by the reviewers of 0.25°x0.25° would lead to a factor 4 increase in the data volume, which we could no longer handle.
- Even though we "only" collect data for a few years, since we are retrieving 50 members twice a day, each with a lead time of 15 days, we still end up with an impressive dataset covering 2*50*15=1500 days of forecast data for each day we download.

Below we provide a one-to-one response to all points raised by the reviewers. The reviewers' comments are in black and our replies in blue. Please note that we always refer to the lines in the updated, revised manuscript (document without track changes). We supplement this document with a latexdiff-pdf showing changes since the last version of the manuscript.

**Reviewer 1**

Recommendation: major revisions

Summary: In their article, [Katharina Hartmuth et al.] present a novel method to assess the forecast skill of the ECMWF ensemble in predicting extreme weather events associated with Mediterranean cyclones. The first part of their study explains the methodology and illustrates it with three case studies of impactful Mediterranean cyclones. The forecast skill is evaluated based on the ability to predict the occurrence of extreme precipitation and extreme surface winds (both defined by exceedance of the local 99th percentile).

The paper is overall clear and well-structured. Particular attention has been given to the description of figures, which makes them especially pleasant to read. In the following, you will find some suggestions, mostly minor, that may help improve the paper.

We thank the reviewer for this positive overall evaluation and for emphasizing the figure descriptions, because indeed we invested a lot in the design of the figures and their description.

Major revisions

0.5°, 6 h: As discussed in your conclusion, using 0.5° may be limiting, especially if you look at small objects (such as medicanes). Also, 6 h is coarse for the Mediterranean, where storms evolve quickly. If the work is not too big, I strongly encourage you to take full advantage of the available resolutions. Another way (if increasing resolutions is not possible) could be to use products like "accumulation of precipitation within the 6 h" or "maximum wind gust within the 6 h" if they are available.

We start with the second part of the comment. Thank you for bringing this up, because we already use 6-h accumulated precipitation and maximum wind gust within 6 h as part of the standard output of the IFS ensemble. We will explain this more clearly in the revised version. With regard to the temporal and spatial resolution: output from the IFS ensemble is available "only" every 6 h for the entire 15-day forecast range (higher-frequency output would be available during the first six days (every 3 h) and the first 90 h (every 1 h), but using an inhomogeneous temporal resolution would make our study even more complicated. Regarding the spatial resolution, this is clearly a compromise. As outlined in the general remark above, doing our data processing at a higher resolution (e.g., 0.25°) would render it unfeasible. Furthermore, although we retrieve and evaluate the forecast data on 0.5°, some of the improved information of the higher native resolution with which the forecast has been run should still be retained in our coarser dataset.

Radius of 400 km: While this radius is adapted for medicanes, it is not for strong winds associated with extratropical cyclones, even in the Mediterranean. I suggest using a circle of

1000 km around the cyclone centre, or (maybe better) using isolines of pressure that you utilised already to capture cyclones. Also, if you want to avoid attributing high wind to weak lows, you could use a threshold on the minimum central pressure.

Note that we only request that an extreme object overlaps with the circle with a radius of 400 km around the cyclone centre, i.e., we request the closest point of the extreme object to be within 400 km from the centre, but we also consider extreme precipitation and winds much farther away from the centre. This can be nicely seen already in Fig. 2, where the panels show a domain size of about 2000 km x 2000 km. We also thought of using a "dynamic size" based on the isolines of pressure, as mentioned by the reviewer, but then decided to keep certain elements of the method simple, also because then the results are more reproducible by others.

Percentile calculated each season: I do not think that this is relevant when looking at impacts. Indeed, high wind or precipitation do not impact differently following the season but following their strength. I encourage you to recalculate the results based on a fixed threshold for the whole year. Also, generally the 98th percentile has been used for wind gusts [Klawa and Ulbrich (2003)], as it was shown to fit well the observed losses. Finally, you could use the so-called Storm Severity Index to draw conclusions on the prediction of the impacts.

Thank you for your suggestions. We agree that when looking at impacts, annual percentiles might be the best option. However, since one of our long-term goals is to compare the predictability of high impact cyclones across different seasons, we introduce a seasonal threshold in this study. We will clarify this aim in the revised manuscript.
Regarding the percentile itself, we argue that using the 98$^{th}$ percentile is as subjective as using the 99$^{th}$ percentile. Using the Storm Severity Index is a great idea which we will keep in mind for possible future studies.

"Probability": The word probability is sometimes unclear throughout the manuscript. You could reword it in the text when it is relevant, like "proportion of members". The terms "conditional" and "unconditional" probabilities are also adding complexity that may be avoided. This point is a detail, but I think the manuscript could gain clarity with slight modifications.

We clarified the use of the term "probability" in L190f of the revised manuscript. Furthermore, we avoid using the terms "conditional"/"unconditional" but instead only add a footnote on page 24 of the revised manuscript, where we refer to the supplementary figures showing conditional probabilities and briefly explain how those are calculated.

Different operational cycles of IFS: You plan to use different cycles of the IFS (successive improvements of the model) for your statistical analysis. I suggest either using reforecasts

instead or clearly justifying how the use of different cycles will—or will not—affect your results.

Thank you for bringing up this point. Indeed, the cycle updates could affect the statistical analysis planned for our part 2 paper, but this is unavoidable when focusing on the quality of operational forecasts. We assume that predictability differences between individual cyclones are generally much larger than systematic differences between IFS ensemble cycles during the last few years – a hypothesis which we aim to shed light on in the statistical part 2 of this study. Doing a similar study with reforecasts would be of course very interesting – but different from the study we would like to perform with operational ensemble forecasts.

Minor suggestions

Title: The title could be more concise and clearer, maybe something like: Forecast uncertainty of high-impact weather associated with Mediterranean cyclones. Part 1: Method and case studies. Also, the term "forecast uncertainty" may be replaced by "predictability" in your case.

Thank you for this suggestion. We changed the title to "Predictability of high-impact weather associated with Mediterranean cyclones in ECMWF ensemble forecasts. Part 1: Method and case studies."

General comment: Using the passive voice may provide better objectivity.

We think that active voice can be more engaging for the readers. Since active voice is further more and more encouraged in scientific writing (see for example here https://press.uchicago.edu/ucp/books/book/chicago/W/bo15288825.html), we prefer to keep certain parts with active voice.

General comment: A figure illustrating the value of the local 99th percentile of precipitation and wind gusts would be very enjoyable.

We added the following figure to the supplement of the revised manuscript:

[Figure]

Figure R1: Seasonal 99th percentile in (a,b) SON and (c,d) DJF of (a,c) 6-h accumulated $P$ ($P_{99}$, in mm 6 h$^{-1}$) and (b,d) maximum gust within a 6-h period $G_{10}$ ($G_{10\ 99}$, in m s$^{-1}$).

Introduction

L. 4: "ECMWF" not defined at this point.

Thank you, we now use the long-form version instead of the abbreviation in the abstract.

L. 5: "We apply … to attribute Mediterranean cyclones to events of extreme." Is it not the opposite? (We apply … to attribute extreme events to Mediterranean cyclones).

Rephrased to "We apply … to attribute events of extreme precipitation and surface winds to Mediterranean cyclones".

L. 25: The whole sentence could be split into two parts for clarity ('Jans`a … cyclones', 'surface' … '2024)'.

Split the sentence in two parts as suggested.

L. 36-37: How can you say that the forecast skill of the position is higher than for the intensity since it is based on two different parameters?

We cite Froude et al. (2007a) here, which write in their conclusion "The skill in predicting the position of extratropical cyclones is significantly higher than that for the intensity."

We agree that the wording might be confusing when comparing different parameters with different units, which is why we added quotation marks to the quote in the revised manuscript.

L. 59-64: Could be reworded to shorten the sentences and prevent the repetition of "such".

Thank you, we rephrased this paragraph in the reviewed manuscript.

L. 66-80: This part should be made more concise and clearer. For example, the methodology presented in the "Discussion and Conclusions" is easier to follow. The word "whether" line 70 could be replaced, as uncertainty does depend (to a lesser or greater extent) on the following parameters.

We rephrased and clarified this paragraph in the revised manuscript (L69-87).

L. 75-77: "For instance, … 1000 individual forecasts." is unnecessary.

Given that the scope of technical effort for this study has not yet become fully clear, we actually think it might be necessary to keep this sentence to illustrate not only the size of the "machinery" behind, but also the relevance of focusing on the data & method in this part 1 of our study.

L. 75: ERA5 not defined at this point.

We rephrased the sentence.

L. 79: "This first part ... (1) and parts of (2) and (3)." May be reworded.

We rephrased this sentence as we clarified the paragraph.

L. 82: ERA5 not defined at this point.

See reply to comment above.

Data and method
L. 89-90: Passive voice may be more appropriate here.

See reply to general comment above.

L. 98: Replace "Reduced sea level pressure (SLP)" by "Mean Sea Level Pressure (SLP)"

Changed as requested.

L. 107: Replace "active" by "operational"

Changed as requested.

L. 112: "data is downloaded every 6 h": Not very clear. This whole sentence could be included at the beginning of the section. "ECMWF ensemble consists of 50 members initialised twice per day. For each base time, a forecast output is available every 6 h up to a maximum lead time of 15 d."

Thank you for the suggestion, we added the following after the first sentence of the section: "For each initialization time, a forecast output is available every 6 h up to a maximum lead time of 15 d."

L. 137: Can we be sure it is the same cyclone? Does it happen frequently? If yes, maybe a criterion based on the SLP value may be helpful.

We investigated the occurrence of such cases which is dependent on the longevity of the cyclone (for a long-lived cyclone such as Storm Daniel, the occurrence of two matching tracks in the same ensemble member is more frequent, since there are more timesteps where our spatio-temporal criterion can be fulfilled and, thus, matching tracks can be found). The following table shows the number of matched cyclones for each Storm for different lead times. The number of members in which more than one matching track is found is given in brackets.

| Storm | 96h / 90h | 84h / 78h | 72h / 66h | 60h / 54h | 48h / 42h | 36h / 30h | 24h / 18h | 12h / 6h |
|---|---|---|---|---|---|---|---|---|
| Denise | 29 (0) | 34 (1) | 36 (0) | 34 (0) | 40 (1) | 45 (0) | 48 (1) | 50 (1) |
| Jan | 10 (0) | 10 (0) | 9 (1) | 26 (1) | 31 (3) | 41 (10) | 47 (17) | 50 (6) |
| Daniel | 27 (2) | 33 (5) | 38 (11) | 49 (10) | 50 (7) | 50 (16) | 50 (7) | 50 (8) |

In a next step, we analysed the nature of these "double track" occurrences as well as the behaviour of our merging algorithm:

- For Storm Jan, all "double track" occurrences, without exception, feature the case of a first track that matches the earlier phase of Jan (genesis over the North Atlantic, transition into the Mediterranean) and a second track that matches the later phase (slowing down over Sardinia/Italy). Both tracks are in almost all cases merged by our algorithm such that the resulting track does not diverge much from the ERA5 track for more than one timestep (see Fig. R2a,c).

- For Storm Daniel, a similar behaviour of our merging algorithm is shown with two tracks representing the earlier and the later stage of the cyclone. Again, the merging works well in most cases (see Fig. R2b,d).

For a few cases, the merging creates "spikes" in the final track since Track 1 is already diverging from the ERA5 track, while Track 2 did not appear yet (see for example Fig. R2a,d). To evaluate if this is problematic, we applied another merging criterion to discard any trackpoints that have – although being part of a matching cyclone – a distance > 1000km to the respective ERA5 trackpoint to investigate the effect of such "outliers" on our cyclone-centered composites. Such a criterion does not result in any differences in our results which confirms that we do not include extreme surface weather that cannot be attributed to the Storm in ERA5 in these cases. Since we only look at events where P/G10 exceeds the 99[th] percentile and as those outlier trackpoints are still relatively close to the actual cyclone, they do not contribute to our statistics in any way (and furthermore, are within the range of other "matching tracks" in members featuring only one track).

[Figure]

Figure R2: Cyclone tracks for Storm (a,c) Jan and (b,d) Daniel at different lead times. Green lines show members with one track and red lines show merged track of members containing two tracks. Light grey lines show parts of ENS tracks before/after the existence of a track in ERA5 (black line).

To conclude, depending on the longevity of the storm, a "double track" case can occur in up to 30% of all members, where multiple tracks represent different cyclone stages. However, a detailed analysis of these cases shows that our merging algorithm works very well in combining several tracks to one track that is reasonably similar to the ERA5 track.

We added some more details about the merging algorithm in Sect. 2.3 of the revised manuscript and added another panel to Fig.1 to improve the schematic explanation of the mechanism.

$\theta$: Usually, $\theta_e$ or $\theta'_w$ are better adapted to look at fronts.

We agree, but since the identification of fronts is not a key part of our study, and θ is relevant for visualizing baroclinicity, we keep θ.

L. 158: "probability of such object". Not very clear, is it a probability of occurrence? If yes, please clarify in the text.

Yes, it is indeed a probability of occurrence of an extreme object. We clarified this in the revised manuscript (L190f).

Fig. 3, 5, 7: "number of overlapping objects" refers to extreme events detected in X members. Maybe the legend could be reworded to enhance clarity. The stars are too small.

Figures 3, 5, and 7 are discussed in Sect. 3, which is not yet looking at ensemble data but exclusively at ERA5 data of the three case study storms. The "number of overlapping objects" refers to the number of timesteps along the cyclone track where at a certain grid point an object of either extreme precipitation or surface wind gusts is detected. We clarified this in the revised manuscript in L221f as well as in the figure caption and now use a new colorbar title ("track points with extreme object"). We increased the size of the stars.

Case study overview
Table 1: "Maximum intensification" may be in [hPa / 12 h] for the three cases to enhance clarity and to fix the given unit.

In the original manuscript we were showing the strongest intensification for differing time periods (12 h for Storm Denise, 18 h for Storms Jan and Daniel). To avoid confusion, we adjusted the table such that we only consider the strongest deepening within 12 h and added the correct unit.

Fig. 4, 6, 8: I suggest slight modifications to these nice-looking figures. White coastlines a bit thicker, grey contours a bit thicker, black star larger, TH [K] to θ[K]. The areas are also difficult to see with the colours (blue on blue and orange on orange).

Thank you for all these suggestions! We implemented all the suggested modifications to improve the figures in the revised manuscript.

L. 206: I do not think it "passed across the Pyrenees", at least for the storm centre. It seems that it remains on their North side.

Changed to "passed along the northern edge of the Pyrenees".

L. 206: by 10 hPa / how much time?

The storm intensified by 10 hPa in 18 hours. Clarified in the revised manuscript.

L. 208: "as a medicane by EUMETSAT": I would be careful with this sentence, since the actual definition given by [Miglietta et al., 2025] is more precise. A consensus did not exist at the time EUMETSAT did that categorisation for a storm evolving in the middle of winter. In particular, "the warm core" was not formally detected. Also, if you choose to keep the sentence, the reference given should be corrected in L. 482 (northern Atlantic? + double https)

Thank you for pointing this out. We deleted this part of the sentence in the revised manuscript.

ECMWF forecast performance
L. 246: I would replace the word "performance" by "skill" in the whole manuscript since we compare to ERA5.

The term "skill" has a very specific meaning in forecast verification, it refers to the forecast performance relative to a reference (e.g., climatology). Not every score is a skill score. Since we are here less specific about the forecast measure, we prefer to use "performance".

Fig. 9: The figure was difficult to read, but the description was really useful. However, I suggest two simplifications: replace the word "probability" by "percentage of members", or plot directly the number of members. Replace "time relative to cyclogenesis in ERA5" by something like "Initialization time [h]". The parts after "0" seem not very useful for the analysis and may be removed.

Thank you for these suggestions. We changed the label "probability" to "percentage of members" in the revised manuscript. We prefer to keep the x-axis label, since something like "initialization time" is not accurate enough and could lead to confusion, especially since here it is specifically the time relative to cyclogenesis in ERA5. We further prefer to keep the parts after "0", since they include valuable information for example about the lifetime of the cyclone.

L. 301: "They contain many small panels". Replace by "Each panel represents."

Changed as suggested.

L. 307: "averaged extreme object probability of ENS within the ERA5 object", may be simplified.

Since this sentence can hardly be simplified without this resulting in the lack of relevant information, we rephrased it to "Additionally, Fig. 11 shows the probability of extreme objects in ENS averaged within the ERA5 object ($p_{obj}$)...".

Fig. 10, 12, 13: The figures could be simplified by plotting $t_{fc}$ at only two times (cyclogenesis and mature stage). It would also have the advantage of avoiding a supplementary notation ($t_{cyc}$) while providing useful information. Also, if you modify Fig. 9 as stated earlier, maybe replace ($t_{fc}$) by "Initialization" or "$t_{ini}$" for consistency.

We are aware that Figs. 10, 12 and 13 contain many panels. However, we are convinced that removing some timesteps here would result in the lack of valuable and relevant information. Showing only two timesteps along the cyclone lifecycle would not allow us to show the progression of forecast performance with increasing lifetime of the cyclone. While this information is also included in Fig. 11, relevant information about positional forecast error relative to the size of the extreme object would get lost (which is one of three key aspects found that affect the probability of extreme surface weather objects). We think that although there are many panels, once the reader has understood how the figure "works", the variety of interesting results included by showing all panels as in the first version of the manuscript will outweigh potential confusion or the feeling of being overwhelmed by too many panels.

Fig. 11: If you do modify Fig. 10, 12, 13, consider modifying Fig. 11 by plotting only cyclogenesis and the mature stage. If you want to keep the area size, make it larger. It would be easier to understand, and you can still provide additional information in your part 2 if it is meaningful. And last, maybe the figure is misplaced; I would expect this information after reviewing the 3 case studies.

The point of adding Fig. 11 is to actually include all cyclone timesteps, also the ones that could not be shown in Figs. 10, 12 and 13 to better understand the evolution of the forecast of extreme objects along the cyclones' lifecycles. Indeed, we could include further meaningful information in part 2 of the paper, however, one main aspect of part 1 is to show that using only three case studies already reveals a large variability in the forecast performance of the associated extreme weather objects.

To avoid that the reader has to jump between figure types, we inserted the respective panels of Fig. 11 into Figs. 10, 12 (new: 11) and 13 (new: 12) in the revised manuscript.

Conclusion
L. 366: "three three"

Changed to "the three".

L. 371: This sentence could be discussed a bit more. Indeed, we expect the strongest winds to be located in the southern parts of extratropical cyclones, but in medicanes they tend to occur all around the centre.

We added a more extended version of this sentence to the revised manuscript.

L. 400: Could be discussed for medicanes that are small objects with high precipitation.

We understand that medicanes are fascinating and important, but there are only few of them and our method therefore should be most useful for all Mediterranean cyclones and not specifically for medicanes.

L. 435: It would be also interesting to see how prediction varies depending on cyclone type (baroclinic vs. strongly diabatically impacted).

Indeed, thanks for this good suggestion, which we will keep in mind when designing our part 2 study.

**Reviewer 2**

Recommendation: major revisions

The paper presents a new object-based framework to quantify ensemble forecast skill of extreme surface weather events linked to Mediterranean cyclones, demonstrated through three case studies (Denise, Jan, Daniel). The study finds good predictability at short lead times but high case-to-case variability at longer ranges. This contribution is clearly within WCD's scope and addresses questions of societal relevance.

General assessment

The manuscript is strong and well-prepared. It provides a detailed and systematic methodology, and the case studies are relevant and timely. The figures are carefully prepared and generally of high quality. The paper also addresses societally important questions, given the devastating impacts of recent Mediterranean storms, which underlines its relevance for early warning and risk preparedness. One central issue, however, is that the role of this Part 1 relative to the forthcoming Part 2 is not entirely clear; it would be important to ensure that Part 1 stands on its own with distinct take-home messages.

Thank you for the positive general assessment of our study and we appreciate in particular that you mention the systematic methodology and the quality of the figures. We apologize that the distinction between this paper (Part 1) and Part 2 was not clear. In our view, this first part stands on its own as it introduces a new methodology and illustrates it with the aid of interesting and relevant case studies of Mediterranean cyclones.

Where the other reviewer emphasized technical aspects of resolution and thresholds, which I completely agree and I think would greatly improve the manuscript, my main suggestions will relate to:

- Framing and motivation (why this method, and how it complements existing approaches),
- Balance between method and science (it currently reads primarily as a methodological paper with case studies as illustrations and could be strengthened by drawing out more physical insight),
- Integration of results across cases (a more comparative discussion would make the findings stronger),
- Positioning relative to Part 2 (clarify what this paper stands alone for, and what will only be delivered later).

We will address all these points below.

Overall, the paper is a solid methodological contribution and of clear relevance to WCD, but I recommend major revisions to sharpen the message. To maximize impact the manuscript should: (i) better highlight the scientific insight gained from the case studies, (ii) improve the comparative and interpretive discussion, and (iii) clarify the role of this Part 1 relative to Part 2.

Specific comments

Motivation and novelty: The paper could better explain how the proposed framework advances beyond existing object-based verification methods, and why the cyclone-centred perspective is particularly valuable compared to more classical metrics.

The main novelty, in our view, is that we apply such an object-based approach to a large set of operational ensemble forecast data. To keep the methodology feasible (also potentially for other research groups who would like to apply it to their forecast data), the proposed framework is pragmatic (compromise in terms of resolution, definition of extreme objects, association of extreme objects to cyclones, etc.). We don't claim that this method per se is novel, but its systematic application to ensembles is (to the best of our knowledge). About the cyclone-centred perspective: we think that this approach is complementary to more classical Eulerian metrics. Clearly, if you are interested in forecast quality at a certain location, then the cyclone-centred approach would not be the best choice. However, if we are interested in general, how well the IFS ensemble can predict extreme weather associated with cyclones, then this approach can be very meaningful, as it provides dynamical context. To give an example, our statistical analysis in Part 2 might be able to reveal that forecast uncertainties associated with precipitation extremes are larger along the cold frontal compared to the warm frontal region.

Balance between method and science: The manuscript leans heavily on methodological description, with the case studies serving mainly as illustrations. The paper would be strengthened by drawing out more physical insight from these examples. The scientific insights could be expanded by linking forecast skill more explicitly to storm characteristics (e.g., intensity, lifecycle, baroclinic vs. diabatic influences).

We agree that this paper is more on the methodological side, but we don't see this as a weakness. For the three case studies, we mention their intensity and lifetime, but we cannot draw robust conclusions about the links between forecast performance and cyclone characteristics from three cases. This is something we hope to be able to do in the statistical analysis in Part 2. Quantifying baroclinic vs. diabatic influences is not straightforward, but Fig. R3 – which shows Eady Growth Rate for all three cases – indicates some case-to-case variability. While baroclinicity seems to be less important in the case of Storm Daniel (Fig. R3c,f) and, in particular, its later, more stationary phase, it plays a more important role in the case of Storm Jan (Fig. R3b,e) and even more for Storm Denise (Fig. R3a,d).

[Figure]

Figure R3: Eady Growth Rate (EGR; color, in $h^{-1}$) for the time steps shown in Figs. 4, 6, and 8 in the manuscript. Left column shows time steps for Storm Denise, middle column shows time steps for Storm Jan and right panel shows time steps for Storm Daniel. Cyclone center is marked by black star. Areas of extreme $P$ and $G_{10}$ are shown with blue and orange hatching, respectively.

Comparative discussion of cases: Each storm is described in detail, but the synthesis across cases is limited. A stronger side-by-side comparison would highlight important contrasts (e.g., Storm Jan's poor predictability vs. Denise's higher ensemble detection). A summary table or concise figure could help.

Thank you for this suggestion, we added a summary table in the concluding section.

Scope of Part 1 vs. Part 2: The division between what this paper achieves and what will follow in Part 2 should be clearer, to ensure Part 1 stands on its own as more than a methodological note.

Thanks for this remark, we now describe the specific aims of parts 1 and 2 better in the revised version (L69f).

Assumptions and parameters: Some methodological choices need fuller justification (e.g., attribution radius of 400 km, seasonal percentiles, coarse temporal/spatial resolution). Even if not modified, sensitivity or limitations should be discussed more explicitly. This point has already been noted by another reviewer, and I also consider it essential to address explicitly.

When processing such a huge dataset (see general remark at the beginning of this document), then pragmatic choices must be made. In the revised version, we better highlight these choices and justify them where possible (see for example added paragraph in L185f), but a robust sensitivity study, e.g., with different resolutions is not feasible.

Figures and presentation: Figures are of high quality but often too dense (e.g., Figs. 9-13). Simplification, clearer legends, or moving some panels to supplementary material would improve readability. In addition, some legends (e.g., "number of overlapping objects") could be clarified, and colour/hatching choices made more distinct.

While we agree that a reduction of some figures to less panels might enhance readability, we are convinced that this would go along with a loss of interesting and relevant information (see reply to comment of reviewer 1 on page 10 of this document). We further think that the complexity of these figures will exist anyways – no matter if we show two or three timesteps – but that showing several cyclone timesteps adds another dimension (the importance of the cyclone lifecycle) without adding further information to single panels.

We improved the legends where feasible (see replies to comments of reviewer 1 above) and improved the visibility of the figures in the revised version of the manuscript. We furthermore integrated the panels of Fig.11 into Figs. 10, 12 and 13.

Terminology and clarity: As the other reviewer noted, the use of "probability" for "fraction of ensemble members" may mislead readers; either define this clearly or use alternative wording such as "ensemble fraction." Likewise, conditional/unconditional terminology could be simplified.

Thanks for mentioning that our terminology was partly confusing. We improved the use of the word "probability" in the revised manuscript (L190f) and furthermore avoid using the terms "unconditional/conditional" (also see reply to comment of reviewer 1 above).

Connection to impacts: The discussion could more strongly link the findings to societal relevance, such as implications for early warning systems and the realistic lead times at which reliable forecasts of Mediterranean extremes can be expected.

We understand and appreciate the reviewer's interest in these aspects, but we don't think that we can identify robust conclusions about realistic lead times from three cases only. However, the large variability in predictability we find across the three cases already indicates that there is large case-to-case variability and therefore, most likely, no "easy and general" message for early warning systems. We discuss this a bit more in the revised version, and in Part 2, we expect to be able to provide a more robust answer to this important question, when studying a larger set of events.

With these revisions, the paper will not only present a valuable methodology but also provide clearer scientific lessons from the case studies, making it an excellent contribution to WCD. I therefore recommend major revisions, with the expectation that a revised version could be suitable for publication.

Thank you.

Evaluation according to WCD criteria

In terms of scientific significance, I consider the contribution to be good. The object-based ensemble method represents a useful innovation and is clearly relevant to the study of Mediterranean cyclone predictability. However, as this paper mainly introduces the methodology and demonstrates it through three case studies, its broader scientific impact is currently modest but could be increased with greater emphasis on the insights gained from the case studies.

We agree that the insight from a few case studies is naturally limited. However, since we implemented a rather novel and technically involved methodology, we find it important to carefully introduce our approach and show how it works with the aid of case studies. Without this paper, we cannot do a broader analysis of the predictability of extreme weather conditions related to Mediterranean cyclones in part 2.

Regarding scientific quality, the work is carefully carried out and methodologically sound. The authors describe their approach in detail and apply it systematically. Nevertheless, several assumptions—such as the choice of attribution radius, the use of seasonal percentiles, and the reliance on relatively coarse temporal and spatial resolution—require more justification. These issues do not invalidate the results but leave some uncertainty about their generalizability.

See our answers to the specific comments above.

The presentation quality of the manuscript is overall good. The paper is well structured, written in clear English, and supported by a rich set of figures. At the same time, the manuscript would benefit from some streamlining: certain sections are verbose, some figures are overly complex, and the main messages could be highlighted more strongly. Simplifying figure layouts and focusing the discussion on key findings would enhance readability.

We did our best to simplify aspects that appeared overly complex. However, sometimes, good methods are to a certain degree "complex", and we would not want to hide the details. Therefore, we are reluctant to do strong simplifications.

**Reviewer 3**

Recommendation: major revisions

This paper evaluates a new method for detecting and characterising high-impact weather systems using objective diagnostics. The approach is applied to reanalysis and ensemble datasets, with the aim of improving the understanding and detection of extreme events related to Mediterranean cyclones. The work is very relevant, as it supports the operational need for new and user-friendly tools to identify hazardous weather in a timely way.

Many thanks for this positive overall evaluation.

Major comments

The analysis currently relies on quite low spatial and temporal resolution. While this is a reasonable starting point, it may not adequately capture small and rapidly evolving systems such as medicanes, particularly in the Mediterranean. If feasible, the authors could consider exploiting higher spatial/temporal resolution data to better resolve extremes. It may be a lot of work and not feasible for this paper, but one good option would be to use the CERRA reanalysis instead of ERA5, as it has much higher temporal resolution and closer to the resolution of the ENS IFS. Also, different IFS model versions are used, and this may affect the final results. You should check if, at least, the ENS horizontal resolution didn´t change (from 18 km to 9 km) between your case studies, as it may give different results. In that case, it may be better to use re-forecasts (hindcasts) to always have a similar model configuration.

We refer to our general remark at the beginning of this document as well as to our reply to comments by reviewer 1 regarding the availability for higher spatio-temporal resolution for this study.

The use of different IFS model versions is almost unavoidable when looking at operational ensemble forecasts. The horizontal resolution of ENS did change in June 2023, which is between case studies 2 and 3. One aim for part 2 of this study is the evaluation of systematic improvements following this increase in resolution, which presumably are comparatively small given the large case-to-case variability in the predictability of individual cyclones.

Using re-forecasts would be undoubtedly interesting, however, it would simply be a different study. We would not be able to download the full model level fields when using re-forecasts (see general remark at the beginning of this document) and, thus, not be able to write part 2 of this paper. Furthermore, reforecasts would have a lower number of ensemble members, which likely reduces the quality of the spread, as well as a lower spatial resolution, which is both relevant for such small-scale extremes.

Figures R4-R6 show the comparison between the ERA5 and the CERRA reanalysis for single time steps during each of our case studies. In each case, the fields of both $P$ and $G_{10}$ are fairly similar given the reduced resolution of ERA5. Most importantly, the identified extreme objects (the black line represents seasonal 99[th] percentile of the respective parameter) are similar between both datasets, which shows that ERA5 – despite its lower spatiotemporal resolution – shows reasonable and meaningful results.

[Figure]

Figure R4: Maps of (a,b) 6-h accumulated $P$ (color; in mm) and (c,d) maximum wind gust within a 6-h period (color, in m s$^{-1}$) for the (a,c) ERA5 and (b,d) CERRA reanalysis at 00 UTC on 22 November in 2022 (cyclogenesis of Storm Denise). Black lines denote areas exceeding the seasonal 99[th] percentile.

[Figure]

Figure R5: Same as Fig.R4 but for 18 UTC on 20 January in 2023 (time of minimum SLP during Storm Jan).

[Figure]

Figure R6: Same as Fig. R4 but for 12 UTC on 10 September in 2023 (time of minimum SLP during Storm Daniel).

The framing of the study objectives is somewhat unclear. As a reader, it is difficult to distinguish which aims apply to this specific paper and which are intended for the companion study.

Thank you for mentioning this. We improved the paragraph about the goal and scope of this specific paper and what we plan to present in part 2 (L69f).

**Review by Michael Schutte**

Recommendation: minor revisions

The study focuses on forecast uncertainty of Mediterranean cyclones with respect to the cyclones' predictability, and their related wind and precipitation extremes. Based on ensemble forecasts from ECMWF, the authors outline a novel methodology to track and assess the representation of cyclone-related weather extremes. Three case studies illustrate the methodology and highlight case-to-case variability in forecast uncertainty of those extremes. Additionally, forecast uncertainty is greater if smaller regions are affected by extremes, during earlier lead times, and if the cyclone is not well captured by the ensembles.

Although my expertise lies not in Mediterranean storms, the methodology appears to yield valuable results. At the same time, the manuscript could be strengthened by considering the following aspects:

- Varying dependence between forecast uncertainty and cyclone lifetime
- A risk for confusing the goals stated in the introduction and those actually addressed by the paper
- Potential differences in bias between ENS and ERA5
- The fairness of comparing predictions of extremes for storms of different lifetimes

These points will be discussed further in the following.

Thank you for reading our paper so carefully as part of the Copernicus Editorial Training Programme and for your helpful suggestions.

Main comments

You note in several places (e.g., l. 11-12 and l. 413-417) that predictions are more uncertain during the earlier stages of the cyclones' lifecycle. While this is consistent with Fig. 11 for storms Jan and Daniel, storm Denise appears to show the opposite behaviour, with uncertainty being highest during the later stages of its lifetime. It may therefore be helpful to acknowledge this varying dependence between forecast uncertainty and cyclone lifetime, rather than presenting it as a general tendency.

Thank you for this important remark. Indeed, Storm Denise shows some differing behaviour which is likely affected by the forecast error caused by the small object size in the later time cyclone stages (see area size numbers in Fig. 11a,b). The uncertainty caused by the area size might "overpower" the effect of an improved forecast for later cyclone stages in this case. To avoid confusion, we added the following note in L451f "We reveal ..., for the cyclone stages with a similar area size of extreme weather objects".

Paragraph in l. 66-80: As a reader it is not immediately clear, which goals you aim to achieve in this study and which you investigate in the second part. It could be beneficial to highlight that these goals apply to both studies together in l. 67. Alternatively, the authors could also mention the goals relevant to this paper at first and outline goals for the second part separately.

The same comment was made by other reviewers. We improved the goal and scope of this specific paper and what we plan to present in part 2 in L69f.

l. 142-146: Considering the different IFS model cycles, how well do these thresholds align between ERA5 and ENS? Is the difference between ENS and ERA5 small enough to be negligible or might it have an effect on how well extreme objects are detected in the ensembles? It could be good to mention this, as it might limit the interpretability of the results otherwise.

Thanks for this question, which clearly addresses one of the main caveats of our study. We don't have enough data from ENS to calculate robust percentiles (in the tail of the distribution). Downloading many years of ENS data would be tedious and not help a lot, because then we would calculate percentiles across strongly differing model versions. Since both ERA5 and ENS are based on (different versions of) the IFS model, our assumption is that it is OK to use the ERA5 percentiles also for ENS. A qualitative a posteriori check for this assumption is to look at extreme P or G10 forecasts at short lead times. And indeed, in Fig. 10, 12, and 13, there is good agreement at short lead times, which validates our approach to a certain degree.

You compare associated extremes and cyclones of different lifetimes. I think the three storms are well chosen, as they also reflect this aspect. However, one might have to be careful when comparing the results of the different storms with each other. For example, in Fig. 9, it is hardly possible to predict a later cyclogenesis for storm Denise, due to its very short lifetime. Thus, the algorithm might miss out any forecast that predicts a storm some days later, opposed to storm Daniel, with potential consequences for the predictability of P and G10. The authors could at least point out this aspect, e.g., in Sec. 4.1.1.

Thank you, we now discuss this aspect better in the revised version where we added a sentence in L420f.

Minor comments

l. 7-8: The authors could define in the abstract what they mean with 'objects of extreme surface weather' These two-dimensional objects are defined nicely in Sec. 2.4., but it might be helpful to provide a short explanation in the abstract, as well.

We added the following sentence to L6f in the abstract: "Thereby, objects of extreme surface weather are identified at grid points that exceed the seasonal 99th percentile of these parameters and matched to cyclones based on their distance to the cyclone center."

l. 12: The authors could add a sentence about implications at the end of the abstract to highlight the relevance of the study's results.

We added the following sentence at the end of the abstract:
"The methodological development and its application documented in this paper provide the basis for a quasi-climatological investigation of the predictability of extreme weather linked to Mediterranean cyclones in a follow-up study."

l. 34 'cyclone speed': As a reader it is not immediately clear what is meant with cyclone speed. Do the authors mean the propagation speed of cyclones, intensification speed/cyclogenesis, or 10-m wind speeds?

Rephrased to "cyclone propagation speed".

l. 106-107: Since you mention the region that the retrieved fields cover, you could also add the coordinates of the domain in parentheses.

Added coordinates in the revised manuscript.

l. 107: Why did you chose 0.5° grid spacing? As you also mention in the conclusions, a higher spatial resolution could be valuable in the case of investigating Mediterranean cyclones, e.g., with respect to cyclone detection and tracking (Aragão and Porcù, 2022).

Please see our general comment about resolution at the beginning of this document as well as our replies to similar comments by the other reviewers.

Fig. 1: I really like this figure to visualize how the cyclone tracks matching algorithm works. Unfortunately, you don't refer to it in the text right now. For example, this could be done at the end of Sec. 2.3.

Thank you for this important remark. We added a reference to Fig. 1 in the revised manuscript.

l. 136-138: How much does this methodological choice increase the number of 'random jumps' in identified cyclone tracks, e.g., in Figure 1, jumping from the first match to the second match and then to the third match? And does this have any potential drawbacks for the analysis of forecast uncertainty? For example, a jump in detected cyclone could result in a sudden change of P or G10 probabilities due to a different location of cyclone centre.

We refer to an in-length answer to a comment raised by reviewer 1 about the matching algorithm. In case of more than one matching track per ensemble member our method actually aims to find the closest and most realistic matching track as opposed to occurring "random jumps" as described in our reply above. We realize that Fig. 1 in the manuscript needs some improvement to avoid the emergence of such an assumption, which is why we added a second panel to the schematic in the revised manuscript. Instead of having such "jumps" as well as including P/G10 fields which can hardly be associated with the cyclone studied in ERA5, the merging algorithm produces continuous tracks that are comparable to members which only contain one track. Furthermore, we find no evidence that P/G10 probabilities are affected but still added a criterion to avoid this when calculating cyclone-centered probabilites.

l. 169-179: The first paragraph of Sec. 2.6 elaborates on the probability of extreme objects, in line with its title. However, the second paragraph justifies why the three storms have been selected as case studies. Thus, it could be beneficial to have a separate section for the second paragraph, e.g., 'Selection of case studies', or revising the title of Sec. 2.6.

Thank you, we added the title "Case study selection" to this paragraph.

Tab. 1: Are the SLP minimum values based on ERA5 data or on observations? Furthermore, you have mentioned socioeconomic impacts of Mediterranean cyclones, so I would suggest adding one more column for only land area/grid points affected by P, and in the same way one column for G10.

The analysis in section 3 is entirely based on ERA5 data as mentioned in L207. Thank you for the suggestion about adding only land grid points affected by P/G10, we added this information to Table 1 in the revised manuscript.

Fig. 9: I really like how you visualize the probability of cyclogenesis in this figure. As especially subpanel (c) is rather long, the readability could be improved if you add thin horizontal lines, e.g., every 20%, in all subpanels instead of having only the 100% line.

We added such lines for better readability in the revised version.

l. 259-263: At lead time –120 h in Fig. 9a, you mention that half of the members are predicting cyclogenesis at the correct time and half too early. However, there is also a small portion coloured in blue, indicating that a few members predict cyclogenesis too late. You could either mention that in the text, or choose a different lead time, e.g., –60 h.

We added a short sentence about the "blue" members in the revised manuscript.

l. 266: Could these differences be linked to different origins, i.e., the North Atlantic vs. the Mediterranean?

In case you are referring to "red vs. green" members at −12 h, cyclogenesis over the North Atlantic can effectively be ruled out, since the cyclone would need to be located close to the Gulf of Genua only 12 h later. Here, the variability in cyclogenesis time is mainly caused by the time interval between forecasts which is 12 h.

Fig. 10: Do the boxes shown in each small subpanel have the same spatial extent as in Fig. 2, i.e., 20° latitude by 20° longitude?

Yes, as described in Sect. 2.6, all panels have an extent of ±10°, which is a spatial extent of 20° latitude by 20° longitude.

Fig. 11a: Here and in Fig. 10a, would it be possible to elaborate shortly, why $p_{obj}$ of P decreases for the −12 h lead time? This seems to be consistent for several times along the lifecycle with exception of earliest time at cyclogenesis.

This is an interesting observation, but we have no convincing explanation for it.

l. 425-427: You could comment on why a higher temporal and spatial resolution might matter for your analysis of Mediterranean cyclones, e.g., an improved detection of precipitation and wind extremes.

We added the following sentence in L469f: "This is especially relevant when looking at small objects of extreme surface weather and due to the fast evolution of storms in the Mediterranean."

Technical comments

l. 89: You could use a comma after 'Then', i.e., 'Then, we present…', to improve readability.

Changed as suggested.

l. 237-238: The sentence 'The position and orientation of the PV streamer is not unlike the one in the early phase of Medicane Zorbas in September 2018' is understandable, but it has some grammatical inconsistencies and could be expressed more clearly. Specifically, 'position and orientation" forms a compound subject, so the verb should also be plural.

We rephrased this sentence to "Both the position and orientation of the PV streamer are similar to the PV streamer present in the early phase of Medicane Zorbas in September 2018" in the revised manuscript.

References

Aragão, L., Porcù, F. Cyclonic activity in the Mediterranean region from a high-resolution perspective using ECMWF ERA5 dataset. Clim. Dyn., 58, 1293–1310 (2022). https://doi.org/10.1007/s00382-021-05963-x

---

## Referee Report (RR1)

**Article review**
"Predictability of high-impact weather associated with Mediterranean cyclones in ECMWF ensemble forecasts.
Part 1: Method and case studies"

**Summary:** In their article, [Katharina Hartmuth et al.] present a novel method to assess the forecast performance of the ECMWF ensemble in predicting extreme weather events associated with Mediterranean cyclones. The first part of their study explains the methodology and illustrates it with three case studies of impactful Mediterranean cyclones. The forecast performance is evaluated based on the ability to predict the occurrence of extreme precipitation and extreme surface winds (both defined by their exceedance of the local 99th percentile).

The authors addressed the questions comprehensively, and the revised version of the manuscript shows significant improvement. Below are some remaining comments and minor suggestions that may help further enhance the final version of the article. Previous comments are shown in italics, the authors' responses are in blue, and, where applicable, supplementary remarks are provided.

**Minor revisions to consider**

*0.5°, 6 h: As discussed in your conclusion, using 0.5° may be limiting, especially if you look at small objects (such as medicanes). Also, 6 h is coarse for the Mediterranean, where storms evolve quickly. If the work is not too big, I strongly encourage you to take full advantage of the available resolutions. Another way (if increasing resolutions is not possible) could be to use products like "accumulation of precipitation within the 6 h" or "maximum wind gust within the 6 h" if they are available.*

We start with the second part of the comment. Thank you for bringing this up, because we already use 6-h accumulated precipitation and maximum wind gust within 6 h as part of the standard output of the IFS ensemble. We will explain this more clearly in the revised version. Thank you for the precision.

With regard to the temporal and spatial resolution: output from the IFS ensemble is available "only" every 6 h for the entire 15-day forecast range (higher-frequency output would be available during the first six days (every 3 h) and the first 90 h (every 1 h), but using an inhomogeneous temporal resolution would make our study even more complicated.

The predictability signal for a cyclone may be extremely weak after a week. I think that your current resolution of 6 h will be a critical limitation in your part 2, probably not if you focus on large PV structures, but very important if you look at smaller scale phenomena. I would strongly encourage keeping the 6 first days with 3 h time resolution, and if it is impossible for the current work, to consider this point for future research within this framework.

Regarding the spatial resolution, this is clearly a compromise. As outlined in the general remark above, doing our data processing at a higher resolution (e.g., 0.25°) would render it unfeasible. Furthermore, although we retrieve and evaluate the forecast data on 0.5°, some of the improved information of the higher native resolution with which the forecast has been run should still be retained in our coarser dataset. I agree with this argument.

*Percentile calculated each season: I do not think that this is relevant when looking at impacts. Indeed, high wind or precipitation do not impact differently following the season but following their strength. I encourage you to recalculate the results based on a fixed threshold for the whole year. Also, generally the 98th percentile has been used for wind gusts [Klawa and Ulbrich (2003)], as it was shown to fit well the observed losses. Finally, you could use the so- called Storm Severity Index to draw conclusions on the prediction of the impacts.*

Thank you for your suggestions. We agree that when looking at impacts, annual percentiles might be the best option. However, since one of our long-term goals is to compare the predictability of high impact cyclones across different seasons, we introduce a seasonal threshold in this study.

I do not fully understand this argument. Would it not be simpler to compare cyclones occurring in both seasons (SON and DJF) using a common threshold? If the intention is to retain season-specific thresholds—which, from my point of view, is debatable—then a clear and comprehensive justification should be provided in the manuscript, similar to the nice one given for the distance threshold (line 178).

Regarding the percentile itself, we argue that using the 98th percentile is as subjective as using the 99th percentile.

I do not fully agree with the argument, since [Klawa and Ubricht, 2003] saw in the 98th percentile a link with insurance losses in Germany. Another argument is that even though your dataset of operational forecasts may contain many members and initialisations, it would not include a large number of different intense cyclones; therefore, the 98th percentile may be more appropriate in a statistical sense.

**Minor suggestions for the revised paper**
The lines given below correspond to those of the revised article (not the track-changes document).

**Title:** The new title is much clearer. You use "high-impact weather" while using "extreme surface weather" or "extreme objects" in every other parts of the manuscript. It may be preferable to unify the terminology throughout the manuscript.

**l.27:** I think the sentence "90/100 of heavy rainfall events in the western Mediterranean are attributed to cyclones" is not exactly what [Jansà *et al.*, 2001] says. "In around 90/100 of all cases of heavy rain in the Mediterranean [...] there is a cyclone centre located within 600 km of the heavy rain site or the MCS centre." It surely exists a convective system within a range of 600 km of a cyclone centre that is not dynamically linked to this cyclone.

**l.41:** Change "poor" by "poorer"

**l.41:** Since [Doiteau *et al.*, 2024] do not use a "skill score" and to be consistent throughout the article, use "performance" here.

**l.61-67:** This paragraph may be better placed after line 48 (or after line 32), which deals with cyclone predictability, rather than after the predictability of extreme events.

**l.66:** "the relevance of such storms for infrastructure and human safety". Should be reworded "*e.g.* the need for accurate predictions of such storms.."

**l.71-76:** "Given..methodology". The introduction was truly pleasant to read until these lines, which are unnecessary and are more appropriately placed in part 2.2. The reader should be able to appreciate the amount of work by reading the paper; therefore, I strongly suggest removing this part.

**l.81:** "quasi-climatological". If you do not plan to study predictability within several decades, keep "multi-year" instead of "quasi-climatological".

**l.84:** Precise the object of "predictability" here (of weather extremes?).

**l.92:** Change the sentence order: "We discuss our results and conclude the study in Sect. 5."

**l.102:** Point 4. is unclear, please reword it. On the opposite, the structure with bullet-points is very easy to read and enjoyable.

**l.105:** ERA5 is already available at 1 h resolution. Reword the sentence, the reader may understand that you interpolated ERA5 every 1 h. Also, since you use 6 h data, it may be appropriate to mention it here.

**l.111:** Check here if the physical parametrisations are also chosen randomly (I think it is the case). If it is exact, include it here along with perturbed initial conditions.

**l.112:** As you said in your first reply, data are available every 3 h the 6 first days. Maybe reword to say that you choose to keep only the 6 h resolution until 15 d.

**l.144 and Fig. 1a.:** "illustrated in Fig. 1a". While Fig. 1b is usefull to get what you did for the merging, the matching is already documented in [Flaounas et al., (2023)] and does not require a figure in this article.

**l.168:** Reword "this is not practical given the challenge" or add a comma.

**l.172:** I do not understand the threshold values here. Are they your 99 th percentile? If it is the case, either reword it to explicitly say it, or remove the sentence. Since the values are quite small, it may not be the case, and if those values are indeed below the 99 th percentile, they are in all cases floored to 0.

**l.185-188:** This part does not improve the scientific objectivity of the paper. I strongly suggest removing it.

**Table 1:** Precise if the SLPmin is from ERA5 or not. Indeed, it seems that the reanalysis underestimates the "true minimum" mean sea level pressure of cyclones.

**l. 210:** "On 22 November" add 2022.

**l. 226:** A sentence could be added to show the coherence between a deeper

cyclone and stronger winds.

**l. 243:** The sentence could be changed to: "Extreme objects were diagnosed only when the storm reached the Mediterranean ..."

**Fig. 9.:** "seasonal climatological cyclone frequency averaged over Mediterranean" I do not understand this. Is it the probability to found a Mediterranean cyclone within the members of the ensemble at any time? Please simplify this sentence or remove the additional information.

**Fig. 10c and d.:** The average probability pobj is still not clear. If I understand, ploc is the proportion of members that found an extreme object at grid point x. Is pobj quantifying "how much" of the predicted object is located within the extreme object of ERA5? Please clarify this point. I would also avoid drawing full lines and dashed ones, since it is not visible in Fig. 13, and since there is already much information to process. If cyclogenesis and cyclolysis refer to ERA5, clarify it in the legend. Finally, it would be very enjoyable to have a scientific "tutorial" on the use of ploc and pobj in the text *e.g.* "The greater ploc the greater/better predicted X", "the greater pobj the greater/better predicted Y".

**l. 463:** remove quasi-climatological.

**l. 471:** "the coverage of different operational cycles" — Here, or alternatively in the Methods section, you could add a brief description of how you intend to quantify the impact of the different model versions on predictability. Otherwise, you may state explicitly your underlying hypothesis, namely that the predictability signal is expected to be stronger than the effects of model improvements.

---

## Author Response (AR2)

*Paper egusphere-2025-4111*

**Quantifying forecast uncertainty of Mediterranean cyclone-related surface weather extremes in ECMWF ensemble forecasts. Part 1: Method and case studies**

by Katharina Hartmuth, Dominik Büeler, and Heini Wernli

We thank two anonymous referees for the feedback on the first revised version of the manuscript. Below, we address further comments by reviewer 1 (in black) with our replies in blue. Previous comments and replies from the first revision round are shown *in italics*. Please note that we always refer to the lines in the updated, revised manuscript (document without track changes). We supplement this document with a latexdiff-pdf showing changes since the last version of the manuscript.

**Reviewer 1**

Summary: In their article, [Katharina Hartmuth et al.] present a novel method to assess the forecast performance of the ECMWF ensemble in predicting extreme weather events associated with Mediterranean cyclones. The first part of their study explains the methodology and illustrates it with three case studies of impactful Mediterranean cyclones. The forecast performance is evaluated based on the ability to predict the occurrence of extreme precipitation and extreme surface winds (both defined by their exceedance of the local 99th percentile). The authors addressed the questions comprehensively, and the revised version of the manuscript shows significant improvement. Below are some remaining comments and minor suggestions that may help further enhance the final version of the article. Previous comments are shown in italics, the authors' responses are in blue, and, where applicable, supplementary remarks are provided.

Minor revisions to consider

*0.5°, 6 h: As discussed in your conclusion, using 0.5° may be limiting, especially if you look at small objects (such as medicanes). Also, 6 h is coarse for the Mediterranean, where storms evolve quickly. If the work is not too big, I strongly encourage you to take full advantage of the available resolutions. Another way (if increasing resolutions is not possible) could be to use products like "accumulation of precipitation within the 6 h" or "maximum wind gust within the 6 h" if they are available.*

*With regard to the temporal and spatial resolution: output from the IFS ensemble is available "only" every 6 h for the entire 15-day forecast range (higher-frequency output would be available during the first six days (every 3 h) and the first 90 h (every 1 h), but using an inhomogeneous temporal resolution would make our study even more complicated.*

The predictability signal for a cyclone may be extremely weak after a week. I think that your current resolution of 6 h will be a critical limitation in your part 2, probably not if you focus on large PV structures, but very important if you look at smaller scale phenomena. I would strongly encourage keeping the 6 first days with 3 h time resolution, and if it is impossible for the current work, to consider this point for future research within this framework.

Thank you for your comment, we agree that a higher temporal resolution would be beneficial, especially when looking at small-scale phenomena. We will consider this point for future research within this framework. For the current work, it is not feasible to manually retrieve this additional data for all cyclones from the MARS archive.
* * *
*Percentile calculated each season: I do not think that this is relevant when looking at impacts. Indeed, high wind or precipitation do not impact differently following the season but following their strength. I encourage you to recalculate the results based on a fixed threshold for the whole year. Also, generally the 98th percentile has been used for wind gusts [Klawa and Ulbrich (2003)], as it was shown to fit well the observed losses. Finally, you could use the so-called Storm Severity Index to draw conclusions on the prediction of the impacts.*

*Thank you for your suggestions. We agree that when looking at impacts, annual percentiles might be the best option. However, since one of our long-term goals is to compare the predictability of high impact cyclones across different seasons, we introduce a seasonal threshold in this study.*

I do not fully understand this argument. Would it not be simpler to compare cyclones occurring in both seasons (SON and DJF) using a common threshold? If the intention is to retain season-specific thresholds—which, from my point of view, is debatable—then a clear and comprehensive justification should be provided in the manuscript, similar to the nice one given for the distance threshold (line 178).

If we applied a common threshold to define objects of extreme P and G10, we would lose the seasonal signal of these parameters. For example, at a grid point that experiences more precipitation in SON compared to DJF (and even more compared to MAM and JJA), a common threshold would in SON lead to larger objects of extreme P than a seasonal threshold, and relatively larger objects compared to DJF. Now comparing SON vs. DJF events would probably lead to the conclusion that objects in SON are larger and, thus, more extreme P occurs, which is – relatively spoken – not true since it just rains more in SON in general. Furthermore, we find that larger objects show a better probability compared to smaller objects. Using a yearly threshold could lead to the misleading result that P extremes in SON are better predicted compared to P extremes in DJF. However, since those SON objects would also include gridpoints with – in a seasonal framework – less extreme P, we could not do a proper comparison of predictability across seasons.

*A second, less technical answer to this question is that, indeed, both approaches (seasonal vs. annual thresholds) have their pros and cons, and we think that, eventually, different stakeholders or even the same stakeholder considering different questions, might favour one approach or the other. We know that, e.g., large-scale P extremes have a strongly differing seasonality in the western vs. eastern Mediterranean (Raveh-Rubin and Wernli, 2015; their Fig. 3) and therefore using annual thresholds would lead to the identification of more cyclones associated with surface weather extremes in SON in the western and in DJF in the eastern Mediterranean. With our approach, we will have similar numbers of cyclones and extremes in both basins in both seasons, which enables more interesting comparisons across seasons and regions.*
* * *
*Regarding the percentile itself, we argue that using the 98$^{th}$ percentile is as subjective as using the 99$^{th}$ percentile.*

I do not fully agree with the argument, since [Klawa and Ulbrich, 2003] saw in the 98th percentile a link with insurance losses in Germany. Another argument is that even though your dataset of operational forecasts may contain many members and initialisations, it would not include a large number of different intense cyclones; therefore, the 98th percentile may be more appropriate in a statistical sense.

*Klawa and Ulbrich (2003) is an important pioneering study relating extratropical cyclones to damage-related losses. However, they looked at Germany, and being aware of the difficulties in robustly relating hazards to impacts across different regions, we doubt that necessarily the same percentile threshold would also link best with losses in the Mediterranean. We therefore still think that either threshold (98$^{th}$ or 99$^{th}$ percentiles) is equally meaningful and for pragmatic reasons we will keep the higher percentile. The 99$^{th}$ percentile is also often used when investigating heavy precipitation events in climate model simulations (e.g., Ban et al., 2021, Climate Dynamics, https://doi.org/10.1007/s00382-021-05708-w).*
* * *
Minor suggestions for the revised paper

The lines given below correspond to those of the revised article (not the track-changes document).

Title: The new title is much clearer. You use "high-impact weather" while using "extreme surface weather" or "extreme objects" in every other parts of the manuscript. It may be preferable to unify the terminology throughout the Manuscript.

Thank you for this suggestion. We are now only using the term "extreme surface weather" instead of "high-impact weather" for consistency throughout the manuscript.

l.27: I think the sentence "90/100 of heavy rainfall events in the western Mediterranean are attributed to cyclones" is not exactly what [Jans`a et al., 2001] says. "In around 90/100 of all cases of heavy rain in the Mediterranean [...] there is a cyclone centre located within 600 km of the heavy rain site or the MCS centre." It surely exists a convective system within a range of 600 km of a cyclone centre that is not dynamically linked to this cyclone.

Thank you for this remark. We rephrased the sentence in L26f: "... found that over 90% of heavy rainfall events in the western Mediterranean occur within 600 km of a cyclone center."

l.41: Change "poor" by "poorer"

Changed as suggested.

l.41: Since [Doiteau et al., 2024] do not use a "skill score" and to be consistent throughout the article, use "performance" here.

Changed as suggested.

l.61-67: This paragraph may be better placed after line 48 (or after line 32), which deals with cyclone predictability, rather than after the predictability of extreme events.

Thank you for this suggestion, this paragraph is now placed at L49f in the revised manuscript.

l.66: "the relevance of such storms for infrastructure and human safety". Should be reworded "e.g. the need for accurate predictions of such storms.."

Changed to: "...is one example that emphasizes the need for accurate predictions of such storms."

l.71-76: "Given..methodology". The introduction was truly pleasant to read until these lines, which are unnecessary and are more appropriately placed in part 2.2. The reader should be able to appreciate the amount of work by reading the paper; therefore, I strongly suggest removing this part.

We disagree that these lines are unnecessary. They emphasize the relevance of focusing on the method in this part 1 which – given the feedback of all reviewers – was not immediately apparent from reading the initial version of the manuscript. However, we agree that this information would be better placed in Sect. 2.2 and moved it there.

l.81: "quasi-climatological". If you do not plan to study predictability within several decades, keep "multi-year" instead of "quasi-climatological".

We changed all "quasi-climatological" back to "multi-year".

l.84: Precise the object of "predictability" here (of weather extremes?).

Rephrased to "...the link between the probability of these extremes and cyclone characteristics such as...".

l.92: Change the sentence order: "We discuss our results and conclude the study in Sect. 5."

Changed as suggested.

l.102: Point 4. is unclear, please reword it. On the opposite, the structure with bullet-points is very easy to read and enjoyable.

Rephrased point 4 to: "Analysis of ensemble forecast probabilities of extreme surface weather objects (in a cyclone-centric framework)"

l.105: ERA5 is already available at 1 h resolution. Reword the sentence, the reader may understand that you interpolated ERA5 every 1 h. Also, since you use 6 h data, it may be appropriate to mention it here.

Changed and extended the sentence by the following: "...forecast validation, featuring a 1-hourly temporal resolution and interpolated to a grid with a spatial resolution of 0.5°x0.5°."

We further added the following sentence: "To make the dataset comparable to ENS (see Sect. 2.2) we only use 6-hourly data in this study."

l.111: Check here if the physical parametrisations are also chosen randomly (I think it is the case). If it is exact, include it here along with perturbed initial conditions.

The ECMWF switched in November 2024 (IFS Cycle 49r1) from the stochastically perturbed parameterization tendency scheme (SPPT) to the stochastically perturbed parametrizations scheme (SPP). Therefore, for the cases discussed in this study, SPP was not yet operational. However, we add information about SPPT and changed the sentence as follows: "... the ECMWF runs 50 medium-range ensemble members with slightly perturbed initial conditions and stochastically perturbed parameterization tendencies during the forecast integration."

l.112: As you said in your first reply, data are available every 3 h the 6 first days. Maybe reword to say that you choose to keep only the 6 h resolution until 15 d.

Reworded the sentence to: "For each initialization time, we keep a 6-hourly forecast output that is available up to a maximum lead time of 15 d."

l.144 and Fig. 1a.: "illustrated in Fig. 1a". While Fig. 1b is useful to get what you did for the merging, the matching is already documented in [Flaounas et al., (2023)] and does not require a figure in this article.

Given the importance of the cyclone track matching for this study, we think that both figures are useful to the reader to fully understand our methods. We agree that the matching is very similar to Flaounas et al. (2023), but readers often appreciate it when they are given the most relevant information in the study itself.

l.168: Reword "this is not practical given the challenge" or add a comma.

Added a comma: "this is not practical, given the challenge".

l.172: I do not understand the threshold values here. Are they your 99 th percentile? If it is the case, either reword it to explicitly say it, or remove the sentence. Since the values are quite small, it may not be the case, and if those values are indeed below the 99th percentile, they are in all cases floored to 0.

Yes, we refer to the 99$^{th}$ percentile here. Rephrased the sentence to: "In a next step, adjacent grid points that exceed $P_{99}$ and $G_{10,99}$, respectively, are defined as extreme surface weather objects...". The abbreviations used have been added in L167f above.

l.185-188: This part does not improve the scientific objectivity of the paper. I strongly suggest removing it.

We added this paragraph since several reviewers commented on these choices and apparently it did not become clear from the previous version of the manuscript why we chose to set up the method this way. Furthermore, it has been the explicit wish of several reviewers to address our choices more explicitly in the revised version of the manuscript.

Table 1: Precise if the SLPmin is from ERA5 or not. Indeed, it seems that the reanalysis underestimates the "true minimum" mean sea level pressure of cyclones.

Added to the table caption: "Characteristics of all three case study storms in ERA5,...".

l. 210: "On 22 November" add 2022.

Added as suggested.

l. 226: A sentence could be added to show the coherence between a deeper cyclone and stronger winds.

Added the following: "Compared to the two other cases, the area affected by extreme winds is more than 3 times larger, which is coherent with Storm Denise showing the lowest minimum central SLP of all three storms (Table 1)."

l. 243: The sentence could be changed to: "Extreme objects were diagnosed only when the storm reached the Mediterranean ..."

Reworded the sentence to: "Extreme objects were diagnosed only when the storm reached the European continent as shown in Fig....".

Fig. 9.: "seasonal climatological cyclone frequency averaged over Mediterranean" I do not understand this. Is it the probability to found a Mediterranean cyclone within the members of the ensemble at any time? Please simplify this sentence or remove the additional information.

As described in Sect. 2.3, "Mediterranean cyclones are identified as cyclones that reach their mature stage, i.e., their minimum central SLP, within a ``Mediterranean box'' extending from 10°W to 40°E and 30°N to 47°N (except for the Bay of Biscay in the northwestern corner). A seasonal climatological Mediterranean cyclone frequency is calculated as the spatial average of the seasonal cyclone frequency at each grid point in this box." In the caption of Fig. 9, we refer to this averaged seasonal cyclone frequency. We simplified the sentence to the following: "Light blue line denotes an averaged seasonal cyclone frequency as detailed in Sect. 2.3."

Fig. 10c and d.: The average probability pobj is still not clear. If I understand, ploc is the proportion of members that found an extreme object at grid point x. Is pobj quantifying "how much" of the predicted object is located within the extreme object of ERA5? Please clarify this point. I would also avoid drawing full lines and dashed ones, since it is not visible in Fig. 13, and since there is already much information to process. If cyclogenesis and cyclolysis refer to ERA5, clarify it in the legend. Finally, it would be very enjoyable to have a scientific "tutorial" on the use of ploc and pobj in the text e.g. "The greater ploc the greater/better predicted X", "the greater pobj the greater/better predicted Y".

As stated in L344f "Figs.10c,d, 11c,d, and 12c,d show the probability of extreme objects in ENS averaged within the ERA5 object (p_obj; averaged over orange and black contour in panels (a) and (b) above)", p_obj is the averaged p_loc within the ERA5 object contour. We added a reference to Sect. 4.2 in the caption of Fig. 10 and clarified that cyclogenesis/-lysis is referring to ERA5 in the figure caption.

We added the following to L335f: "From here on, these spatial probability fields are referred to as local probability $p_{loc}$, whereby a higher value of $p_{loc}$ at a certain grid point represents a higher number of ensemble members predicting extreme objects at this grid point." and furthermore the following in L346f: "... to further condense the information shown in the panels above. The greater $p_{obj}$ at a certain timestep $t_{cyc}$, the better the prediction of the associated extreme object."

l. 463: remove quasi-climatological.

Deleted as suggested.

l. 471: "the coverage of different operational cycles" — Here, or alternatively in the Methods section, you could add a brief description of how you intend to quantify the impact of the different model versions on predictability. Otherwise, you may state explicitly your underlying hypothesis, namely that the predictability signal is expected to be stronger than the effects of model improvements.

We added the following sentence in the method section 2.2 in L111f: "While the consideration of different IFS Cycles is unavoidable for this study, we expect that the predictability signal is generally stronger than systematic differences between the different cycles."